# Quantitative mapping of protein-peptide affinity landscapes using spectrally encoded beads

Huy Quoc Nguyen[1‡], Jagoree Roy[2†], Björn Harink[1†], Nikhil P Damle[2†§], Naomi R Latorraca[3], Brian C Baxter[4], Kara Brower[5], Scott A Longwell[5], Tanja Kortemme[6,7], Kurt S Thorn[4#], Martha S Cyert[2], Polly Morrell Fordyce[1,5,7,8*]

[1]Department of Genetics, Stanford University, Stanford, United States; [2]Department of Biology, Stanford University, Stanford, United States; [3]Biophysics Program, Stanford University, Stanford, United States; [4]Department of Biochemistry and Biophysics, University of California, San Francisco, San Francisco, United States; [5]Department of Bioengineering, Stanford University, Stanford, United States; [6]Department of Bioengineering and Therapeutic Sciences, University of California, San Francisco, San Francisco, United States; [7]Chan Zuckerberg Biohub, San Francisco, United States; [8]ChEM-H Institute, Stanford University, Stanford, United States

*For correspondence:
pfordyce@stanford.edu

[†]These authors contributed equally to this work

Present address: [‡]Genentech, South San Francisco, United States; [§]OSTHUS GmbH, Eisenbahnweg, Aachen, Germany; [#]Zymergen, Inc, Emeryville, United States

Competing interests: The authors declare that no competing interests exist.

**Abstract** Transient, regulated binding of globular protein domains to Short Linear Motifs (SLiMs) in disordered regions of other proteins drives cellular signaling. Mapping the energy landscapes of these interactions is essential for deciphering and perturbing signaling networks but is challenging due to their weak affinities. We present a powerful technology (MRBLE-pep) that simultaneously quantifies protein binding to a library of peptides directly synthesized on beads containing unique spectral codes. Using MRBLE-pep, we systematically probe binding of calcineurin (CN), a conserved protein phosphatase essential for the immune response and target of immunosuppressants, to the PxIxIT SLiM. We discover that flanking residues and post-translational modifications critically contribute to PxIxIT-CN affinity and identify CN-binding peptides based on multiple scaffolds with a wide range of affinities. The quantitative biophysical data provided by this approach will improve computational modeling efforts, elucidate a broad range of weak protein-SLiM interactions, and revolutionize our understanding of signaling networks.
DOI: https://doi.org/10.7554/eLife.40499.001

## Introduction

*In vivo*, rapid regulation of weak, transient protein-protein interactions is essential for dynamically shaping cellular responses. Nearly 40% of these interactions are mediated by 3–10 amino acid Short Linear Motifs (SLiMs) interacting with protein globular domains (*e.g.* SH3, SH2, and PDZ domains) or enzymes (*e.g.* kinases and phosphatases) (*Dinkel et al., 2016*; *Neduva and Russell, 2006*; *Tompa et al., 2014*). The human proteome is estimated to contain more than 100,000 of these SLiMs, many of which are highly regulated by post-translational modifications (PTMs) such as phosphorylation (*Tompa et al., 2014*; *Ivarsson and Jemth, 2019*). As the weak affinities of these interactions ($K_d$ values of ~1 to 500 μM) are often close to the physiological concentrations of the interacting partners *in vivo*, subtle differences in concentrations or interaction affinities can have large effects on the fraction bound and downstream signaling output (*Roy et al., 2007*; *Hein et al., 2017*). Measuring and predicting not only which SLiMs a protein binds but also the affinities of

SLiM-binding interactions is therefore essential for predicting signal strengths within signaling networks, understanding how these networks are perturbed by human disease, and identifying new therapeutic inhibitors (*Uyar et al., 2014*).

Calcineurin (CN), a conserved $Ca^{2+}$/calmodulin-dependent phosphatase, is a prime example of a signaling protein that relies on weak SLiM-mediated interactions for substrate recognition. Although CN plays critical roles in the human immune, nervous, and cardiovascular systems and likely dephosphorylates hundreds of downstream targets, only ~70 are known to date (*Sheftic et al., 2016*). These include the NFAT family of transcription factors, whose dephosphorylation by CN is required for T-cell activation and adaptive immunity (*Jain et al., 1993*). Consequently, CN is the target of the widely used immunosuppressants cyclosporin A (CysA) and FK506. CN dephosphorylates sites with little sequence similarity, instead recognizing substrates by binding to two characterized SLiMs (PxIxIT and LxVP) located at variable distances from the phosphosite (*Roy and Cyert, 2009*). Blocking SLiM binding to CN prevents dephosphorylation without altering its catalytic center: FK506 and CysA prevent LxVP docking, the viral inhibitor A238L blocks PxIxIT and LxVP binding, and the high-affinity peptide inhibitor PVIVIT blocks PxIxIT binding (*Grigoriu et al., 2013*; *Aramburu et al., 1999*). The affinities of PxIxIT motifs determine biological output by specifying the $Ca^{2+}$ concentration-dependence of substrate dephosphorylation *in vivo* (*Müller et al., 2009*; *Roy et al., 2007*). However, the relationship between PxIxIT sequence and CN binding affinity has never been probed outside of the core motif. A comprehensive understanding of PxIxIT-CN binding would allow discovery of novel CN substrates and aid efforts to rationally design CN inhibitors with enhanced selectivity.

Unfortunately, the limited binding interfaces associated with SLiM-mediated interactions result in low to moderate affinities (with typical $K_d$ values in the range of 1–500 uM), high dissociation rates (*Zhou, 2012*; *Dogan et al., 2015*), and a rapid equilibrium (*Gianni and Jemth, 2017*; *Bagshaw, 2017*), complicating experimental efforts to measure affinities. Display-based methods such as combinatorial phage display (*Tonikian et al., 2008*) and ProP-PD (*Ivarsson et al., 2014*; *Sundell and Ivarsson, 2014*) allow screening for binding between a protein of interest and large libraries (up to $10^{10}$) of candidate SLiM-containing peptides. However, these methods typically identify only strong binders, cannot return negative information about residues that ablate binding critical for downstream target prediction in vivo (*Gfeller et al., 2011*), and do not directly probe effects of PTMs. Array-based methods (*Fodor et al., 1991*) allow quantification of binding of labeled proteins to 10s-100s (SPOT arrays) (*Frank et al., 1990*), tens of thousands (*Atwater et al., 2018*), or millions (ultra-high density arrays) (*Buus et al., 2012*; *Forsström et al., 2014*; *Price et al., 2012*; *Carmona et al., 2015*) of peptides chemically synthesized *in situ*, permitting direct incorporation of PTMs or unnatural amino acids at specific positions (*Engelmann et al., 2014*; *Tinti et al., 2013*; *Filippakopoulos et al., 2012*). However, it is difficult to evaluate the yield and purity of peptides in each spot, contributing to false positives and negatives (*Tinti et al., 2013*; *Blikstad and Ivarsson, 2015*). Most importantly, display- and array-based screening methods cannot measure quantitative interaction affinities ($K_d$s or $\Delta\Delta G$s), as these measurements require washes that take the system out of thermodynamic equilibrium and lead to preferential loss of interactions with faster off-rates (*Zhou, 2012*; *Dogan et al., 2015*; *Ivarsson and Jemth, 2019*). As a result, candidate interactions identified through screening methods are typically validated and their affinities quantified via subsequent quantitative, low-throughput methods that require large amounts of material and are labor-intensive (*e.g.* isothermal calorimetry, fluorescence polarization, or surface plasmon resonance) (*Gibson et al., 2015*; *Davey et al., 2017*; *Dinkel et al., 2016*; *Tonikian et al., 2008*). The ability to measure binding affinities for hundreds of protein-peptide interactions in parallel would simultaneously remove this validation and quantification bottleneck and allow quantitative mapping of binding specificity landscapes to improve target prediction *in vivo*.

Here, we present a powerful bead-based technology for quantitatively measuring affinities for many SLiM-mediated protein-peptide interactions in parallel using very small amounts of material. Peptides are synthesized directly on spectrally encoded beads (MRBLEs, Microspheres with Ratiometric Barcode Lanthanide Encoding) (*Gerver et al., 2012*; *Nguyen et al., 2017a*) with a unique linkage between each peptide sequence and a given spectral code. MRBLE-pep libraries can then be pooled and assayed for protein binding in a single small volume before being imaged to identify the peptide sequence associated with each bead and quantify the amount of protein bound. On-MRBLE chemical synthesis allows for precise control of peptide density, incorporation of PTMs at

known locations, and in-line assessment of peptide quality via mass spectrometry to identify amino acids that ablate and promote protein binding with equal confidence. Most importantly, MRBLE polymeric beads have been deliberately engineered to have slow on- and off-rates, thereby slowing dissociation and allowing quantitative measurement interaction strengths for of weak and transient interactions difficult to detect via other techniques. MRBLE-pep provides a critical complement to high-throughput screening approaches, enabling facile downstream screening of candidate motif hits as well as systematic mapping of how individual residues at each position within the motif contribute to binding affinity and specificity.

We use MRBLE-pep to systematically mutate residues at each position within and flanking three previously-characterized PxIxIT sequences and quantify their effects on affinity. We find that flanking amino acids and post-translational modifications play surprisingly critical and previously unappreciated roles in determining affinity and specificity. Through iterative cycles of mutagenesis, *in vitro* affinity measurements, and *in vivo* activity measurements, we map the CN-PxIxIT affinity landscape and identify several PxIxIT peptides of unprecedented binding affinity that could prove useful for future development of additional potent CN inhibitors. This approach can be applied to a broad range of protein-SLiM interactions to map binding energy landscapes, model signaling networks, and identify novel therapeutic inhibitors.

## Results

### MRBLE-pep experimental assay overview

We developed a new high-throughput protein-peptide interaction assay based on spectrally encoded hydrogel beads containing unique ratios of lanthanide nanophosphors (MRBLEs) (*Gerver et al., 2012*; *Nguyen et al., 2017a*). MRBLEs containing a given code are output to a filtered-tip in a 96-well format or two 48–2 mL reaction tubes (*Figure 1A*). These tips or tubes are then transferred to a peptide synthesizer for functionalization and solid phase peptide synthesis (SPPS), thereby uniquely linking peptide sequences with spectral codes (*Figure 1A*) (*Hintersteiner et al., 2009*; *Lam et al., 1991*; *Lee et al., 2014*; *Liu et al., 2002*; *Meldal, 2002*). Following SPPS, MRBLE-bound peptide libraries are pooled, incubated with an epitope-tagged protein of interest and a fluorescently-labeled antibody, washed, and imaged to reveal the peptide sequence associated with each bead (by quantifying lanthanide emissions) and the amount of protein bound (by quantifying fluorescence) (*Figure 1B,C*). Unlike commercially available spectrally encoded beads (*e.g.* Luminex), spectral codes can be 'read' using a single UV excitation source, preserving the ability to multiplex binding measurements using up to three detection antibodies (*Nguyen et al., 2017b*).

### Integrated synthesis quality control ensures production of full-length, correct peptide sequences

Computational prediction of novel CN-binding SLiMs *in vivo* can be improved by including information about residues that ablate binding (*Krystkowiak et al., 2018*; *Kaushansky et al., 2010*; *Gfeller et al., 2011*). To accomplish this, we implemented in-line peptide quality assessment to ensure that any observed non-binding results from a true absence of interaction and not a failure to synthesize the correct peptide. MRBLEs were first functionalized with an acid-labile rink amide linker within the bead core and a non-labile glycine linker on the outer bead shell (*Figure 1—figure supplement 1*) (*Liu et al., 2011*); varying the ratio of extendible to non-extendible glycine linkers in the bead shell allows tuning of displayed peptide density (*Liu et al., 2002*; *Chen et al., 2009*). Next, peptides were synthesized on both linkers via standard Fmoc SPPS. Peptides coupled to the MRBLE core via the acid-labile linker were eluted during the final acid global deprotection step and can be verified via MALDI mass spectrometry. As peptides within the core are inaccessible to large proteins, elution allows sequence verification without reducing binding signal in downstream binding assays. Measured MRBLE lanthanide ratios before, during, and after SPPS remained constant, (*Figure 1—figure supplement 2*), establishing that embedded codes are unchanged by chemical exposures.

Measuring binding affinities from a single pooled assay requires reliable estimates of the effective concentration of each peptide, which could be skewed by sequence-dependent differences in SPPS efficiency. To detect any large differences, a portion of each MRBLE-pep library was biotinylated,

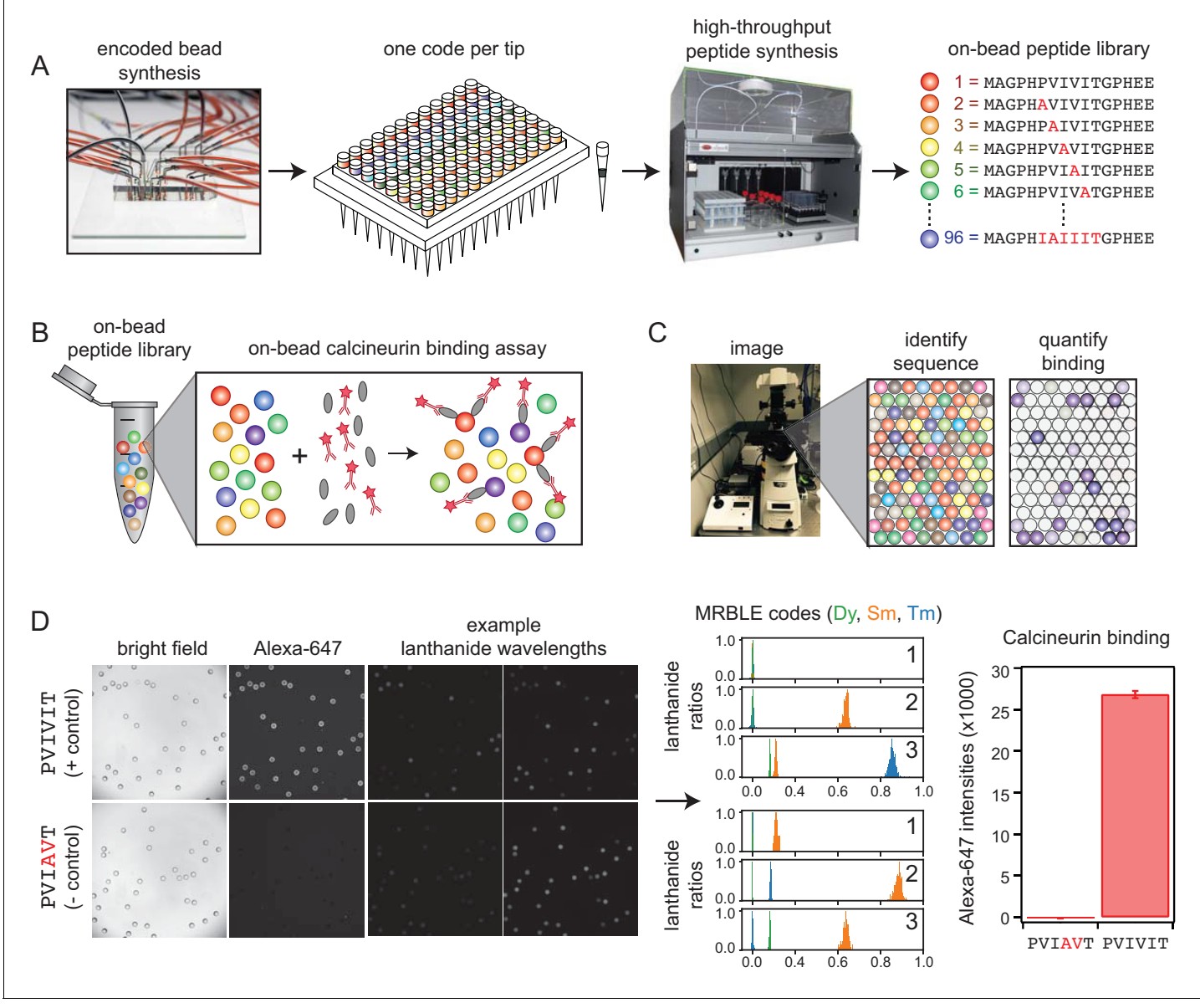

**Figure 1.** MRBLE-pep experimental pipeline for high-throughput measurement of protein-peptide interactions. (**A**) Encoded beads are synthesized in a microfluidic device and output to peptide synthesis tips arrayed in a 96 well plate format with one MRBLE code per tip. Peptides are then synthesized directly on MRBLEs via solid phase peptide synthesis with a unique 1:1 linkage between peptide sequence and spectral code. (**B**) MRBLE-pep libraries are pooled and incubated with an epitope-tagged protein and fluorescently-labeled antibody. (**C**) Following incubation and washing, peptide sequence and amount of bound protein are determined via imaging. (**D**) Example data showing images of MRBLE-pep beads coated with a CN-bound and negative control peptide, measured lanthanide ratios, and bound antibody intensities.

DOI: https://doi.org/10.7554/eLife.40499.002

The following source data and figure supplements are available for figure 1:

**Source data 1.** PVIVIT code ratios.
DOI: https://doi.org/10.7554/eLife.40499.007
**Source data 2.** 'PVIAVT code ratios.
DOI: https://doi.org/10.7554/eLife.40499.008
**Figure supplement 1.** Integrated peptide quality control and quantification.
DOI: https://doi.org/10.7554/eLife.40499.003
**Figure supplement 2.** MRBLE spectral codes are unaffected by chemical reagents required for peptide synthesis.
DOI: https://doi.org/10.7554/eLife.40499.004
**Figure supplement 3.** Streptavidin binding assay reveals any sequence-specific differences in peptide synthesis efficiency.

*Figure 1 continued on next page*

Figure 1 continued

DOI: https://doi.org/10.7554/eLife.40499.005

**Figure supplement 4.** Streptavidin binding assay can be used to determine peptide loading density.

DOI: https://doi.org/10.7554/eLife.40499.006

incubated with labeled streptavidin (DyLight650-SA), washed, and imaged to quantify bound streptavidin. Bound DyLight650-SA intensities were relatively consistent between sequences (*Figure 1— figure supplement 3*) and saturated at ~20 nM DyLight650-SA for ~7100 beads in a 100 µL reaction volume (*Figure 1—figure supplement 4*), establishing a surface density of ~2 × 10$^8$ peptide molecules per MRBLE.

## MRBLE-pep yields quantitative measurements of calcineurin binding affinities

Measuring CN-PxIxIT binding affinities represents a demanding test for a new protein-peptide interaction assay, as CN binds PxIxIT peptide with moderate-to-weak affinities (*Figure 2A*) via surface interactions (*Figure 2B*), and prior literature affinity estimates for a given substrate have varied over 50-fold (*Figure 2A*, NFATc1 and RCAN1). To determine MRBLE-pep assay sensitivity and resolve discrepancies in prior measurements, we performed a pooled assay in which MRBLEs bearing 10 peptides (each synthesized on three spectral codes) were incubated with varying concentrations of His-tagged CN complexed with fluorescently-labeled anti-His antibody. These 10 peptides (the 'Triplicate low' dataset) represent all calcineurin-binding SLiMs for which affinities have been measured, including the known NFATc1, NFATc2, AKAP79, and RCAN1 natural CN-interacting PxIxIT binding sites (*Figure 2A*; *Supplementary file 1*) (*Aramburu et al., 1998*; *Grigoriu et al., 2013*; *Li et al., 2007*; *Mulero et al., 2009*), a set of 5 PVIVIT peptide mutants previously characterized via competitive fluorescence polarization assays (*Aramburu et al., 1999*; *Li et al., 2007*), and a scrambled negative control sequence. Consistent with previous observations, the high-affinity PVIVIT and PVIVVT variants showed strong binding, a scrambled peptide showed no binding, and the PVIAVT and PVIVIN variants showed low to intermediate binding (*Figure 2C*, *Figure 2—source data 1*), with consistent results for the same peptide sequence synthesized on MRBLEs with different codes. NFATc2 and AKAP79 showed measurable binding and were therefore selected for further systematic mutagenesis. Although MALDI mass spectrometry confirmed successful synthesis of both RCAN1 and NFATc1 full-length peptides, binding was near the limit of detection in this assay (*Figure 2C*).

To determine binding affinities for each peptide, we globally fit all data from a given assay to a single-site binding model with the assumption that while individual $K_d$ values may differ between peptides, the stoichiometry of binding remains constant (see Materials and methods). Although these measurements take place out of equilibrium due to the need to wash beads prior to imaging, we specifically engineered MRBLEs to have very slow on- and off-rates (with measured rates on the order of hours) (*Figure 2—figure supplement 1*, *Figure 2—figure supplement 2*), allowing measured intensities to approximate equilibrium values to yield apparent dissociation constants (*Figure 2C*, *Figure 2—figure supplement 3*). Slow on- and off-rates have previously been observed for hydrogel particles and are known to scale with particle radius, as proteins initially encounter and bind bead-bound ligands and then slowly diffuse into the particle via iterative dissociation and rebinding events (*Shapiro et al., 2012*). Resultant apparent $K_d$s largely agree with previously published values (*Figure 2D*) and establish that MRBLE-pep can resolve differences in weak affinities spanning from ~0.20 to ~50 µM (a dynamic range > 2 orders of magnitude, comparable to SPR and FP; *Jiang and Barclay, 2009*). MRBLE-pep apparent $K_d$s slightly underestimate affinities relative to published values while preserving rank order and relative differences; therefore, we provide relative affinity differences ($\Delta\Delta G$) in all subsequent analyses, which are unaffected by this shift.

Examination of known CN PxIxIT motifs (NFATc1, NFATc2) and a viral inhibitor (A238L) suggested a potential preference for positively charged (R, K) or hydroxyl (S) residues in the degenerate PxIxIT position 2 (*Figure 2A*). To evaluate assay reproducibility and whether these substitutions enhance binding, we measured binding to these same 10 peptides plus an additional 3 PVIVIT variants (PSIVIT, PRIVIT, PKIVIT) ('triplicate high') (*Figure 2E*, *Figure 2—source data 2*). These data

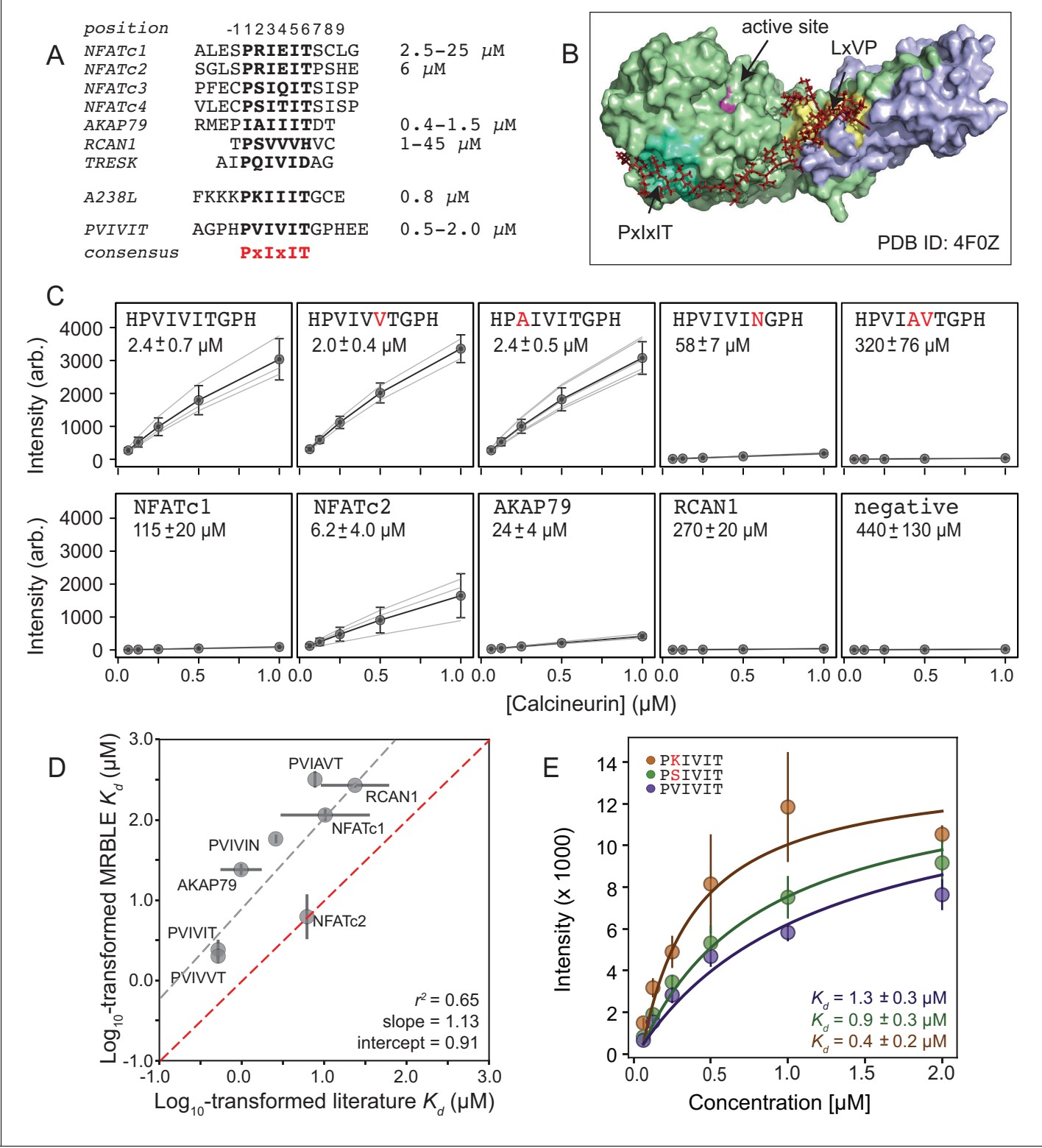

**Figure 2.** MRBLEs allow quantitative measurement of CN-SLiM interactions in high-throughput. (A) Sequence alignment for previously identified CN PxIxIT binding sites (see *Supplementary file 1* for references). (B) Crystal structure showing CN heterodimer bound to the A238L inhibitor including both PxIxIT and LxVP binding sites (PDB ID: 4F0Z). (C) Concentration-dependent binding for CN interacting with either PVIVIT variants (top), natural PxIxIT peptides (bottom), or a scrambled PxIxIT motif (bottom). Each peptide was synthesized on three different MRBLE codes (light grey lines); affinities were determined from global Langmuir isotherm fits (black) to median values at each concentration (grey circles). (D) MRBLE-derived and

*Figure 2 continued on next page*

*Figure 2 continued*

literature reported $K_d$ values for various CN-PxIxIT interactions. Grey solid line indicates linear regression between data sets; dotted line indicates expected 1:1 agreement. (E) Langmuir isotherm fits to concentration-dependent binding behavior for CN interacting with candidate high-affinity PVIVIT core variants.

DOI: https://doi.org/10.7554/eLife.40499.009

The following source data and figure supplements are available for figure 2:

**Source data 1.** 'Triplicate low' affinity measurements.
DOI: https://doi.org/10.7554/eLife.40499.013

**Source data 2.** 'Triplicate high' affinity measurements.
DOI: https://doi.org/10.7554/eLife.40499.014

**Figure supplement 1.** Time-dependent measurements of calcineurin binding demonstrating that calcineurin-MRBLE on-rates are slow.
DOI: https://doi.org/10.7554/eLife.40499.010

**Figure supplement 2.** Dissociation rate of calcineurin over a time period of 13–18 hr after reaching equilibrium.
DOI: https://doi.org/10.7554/eLife.40499.011

**Figure supplement 3.** 'Triplicate high' library of reported PVIVIT and natural variants.
DOI: https://doi.org/10.7554/eLife.40499.012

reproduce previously observed trends and additionally suggest that the PKIVIT peptide binds slightly more strongly than PVIVIT; therefore, we also selected PKIVIT for additional systematic mutagenesis.

## High-throughput MRBLE-pep mapping of the CN-PxIxIT binding energy landscape

Using this new assay, we experimentally mapped portions of the CN-PxIxIT binding energy landscape by measuring CN binding to two MRBLE-pep libraries for each of the PVIVIT, PKIVIT, NFATc2, and AKAP79 sequences containing systematic mutations in either the 'core' (positions 1–6) or 'flanking' PxIxIT residues (positions −1 and 7–9) (368 peptides total) (*Figure 3A*, *Figure 3—figure supplement 1*, *Figure 3—figure supplement 2*, *Figure 3—figure supplement 3*, *Figure 3—figure supplement 4*, *Figure 3—figure supplement 5*, *Figure 3—figure supplement 6*, *Figure 3—figure supplement 7*, *Figure 3—figure supplement 8*, *Figure 3—source data 1*, *Figure 3—source data 2*, *Figure 3—source data 3*, *Figure 3—source data 4*, *Figure 3—source data 5*, *Figure 3—source data 6*, *Figure 3—source data 7*, *Figure 3—source data 8*). As the P and I residues in positions 1, 3, and 5 are known to be strongly conserved across substrates, we prioritized mutagenesis at other, less conserved positions. The optimal number of peptides to screen per assay depends on the competing effects of ligand depletion and competition, the fraction of peptides expected to bind, the range of interaction strengths, and the statistical robustness of each measurement (*Pollard, 2010*). To reduce error and maximize the ability to resolve subtle differences, we profiled 48 peptides per reaction, with ~100 beads per sequence in each assay. To ensure that measured binding resulted from a true CN-PxIxIT interaction and not nonspecific binding, we repeated all experiments at a single, high concentration (250 nM) using a CN N330A/I1331A/R332A mutant defective in PxIxIT binding (CN NIR) (*Li et al., 2004*) and labeled antibody alone (*Figure 3—figure supplement 9*, *Figure 3—figure supplement 10*, *Figure 3—figure supplement 11*, *Figure 3—figure supplement 12*; *Figure 3—source data 9*, *Figure 3—source data 10*). Several peptides were strongly bound by anti-His antibody but did not show binding when the antibody was pre-incubated with His-tagged CN; peptides strongly bound by the CN NIR mutant (*e.g.* PRIRIT) were removed from downstream analysis. All PVIVIT, PKIVIT, and NFATc2 libraries and the AKAP79 'core' mutation library were fit by a single-site binding model, permitting direct measurement of $\Delta\Delta G$ for each substitution. As measured intensities for the AKAP79 'flank' mutation library never reached saturation, MRBLE-pep measurements provide only qualitative estimates of affinity effects.

To visualize the binding affinity landscape of CN-PxIxIT interactions, we generated graphs showing the relative change in affinity upon substitution to each mutant amino acid at each position (*Figure 3B*, *Figure 3—figure supplement 13*, *Figure 3—figure supplement 14*, *Figure 3—figure supplement 15*, *Figure 3—figure supplement 16*, *Figure 3—figure supplement 17*, *Figure 3—figure supplement 18*, *Figure 3—figure supplement 19*, *Figure 3—figure supplement 20*, *Figure 3—figure supplement 21*); information at all positions can be combined to produce a heat map of relative changes in binding energies ($\Delta\Delta G$) (*Figure 3C*) or scaled logos (*Figure 3—figure supplement*

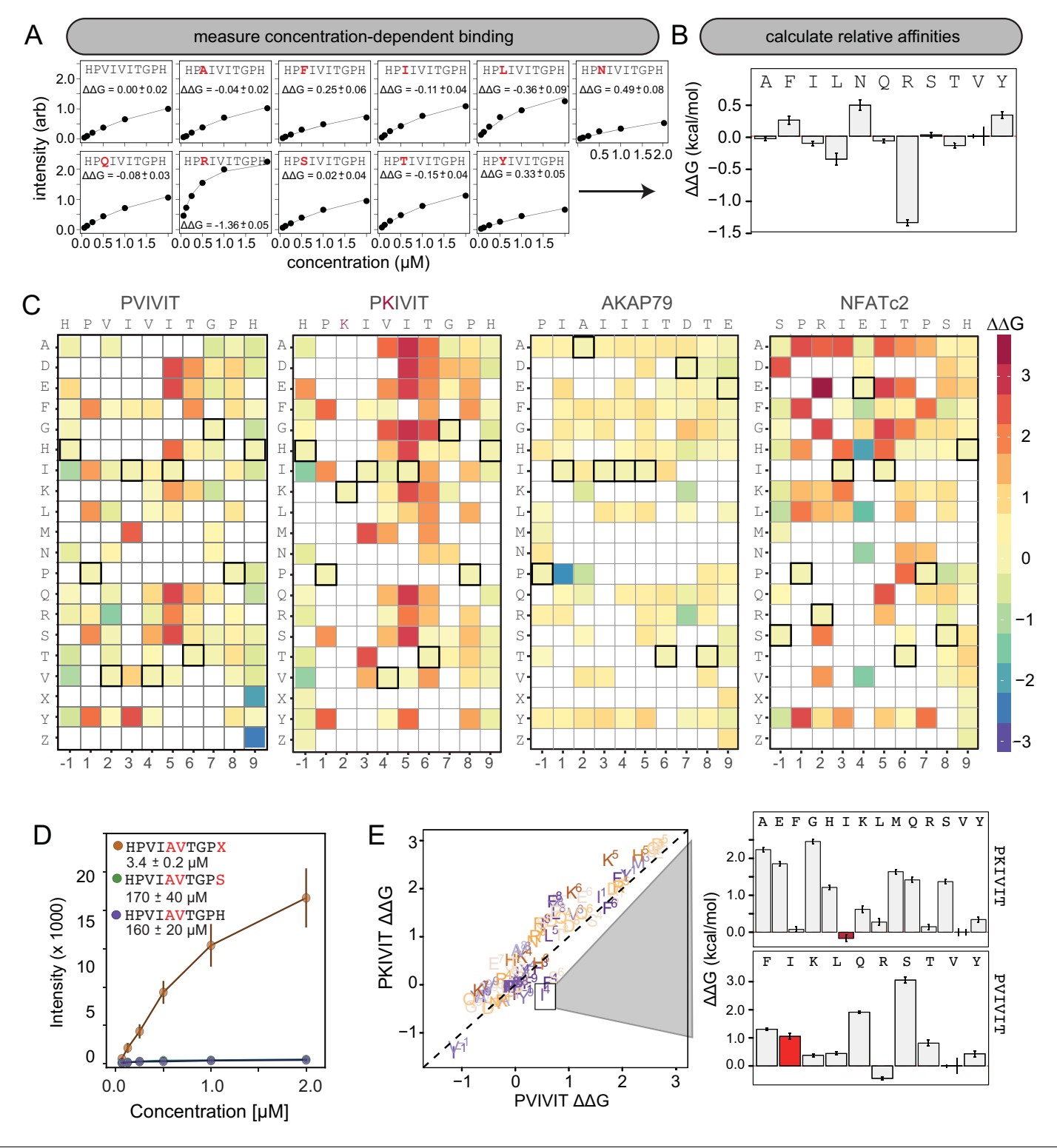

**Figure 3.** MRBLE-pep mapping of the CN-PxIxIT binding energy landscape. (**A**) Langmuir isotherm fits to concentration-dependent binding behavior for CN interacting with PVIVIT variants containing systematic mutations within position 2. (**B**) Relative change in binding affinity for individual substitutions at position 2 (as compared to WT). (**C**) Heat map showing relative affinities per substitution per position for 4 PxIxIT motifs. (**D**) Langmuir isotherm fits to concentration-dependent binding behavior for CN interacting with a medium-affinity PVIVIT variant containing the wild type histidine, a serine, and a phosphoserine at position 9. (**E**) Comparison of the effect on affinity for a particular substitution at a particular position in the PKIVIT motif

*Figure 3 continued on next page*

*Figure 3 continued*

(y axis) vs. the PVIVIT motif (x axis). Inset shows relative effects on affinity for various substitutions at the V4 position in PKIVIT (top) and PVIVIT (bottom), highlighting differential effects of an I substitution.

DOI: https://doi.org/10.7554/eLife.40499.015

The following source data and figure supplements are available for figure 3:

**Source data 1.** Concentration-dependent binding data for PVIVIT core library.
DOI: https://doi.org/10.7554/eLife.40499.039

**Source data 2.** Concentration-dependent binding data for PVIVIT flank library.
DOI: https://doi.org/10.7554/eLife.40499.040

**Source data 3.** Concentration-dependent binding data for PKIVIT core library.
DOI: https://doi.org/10.7554/eLife.40499.041

**Source data 4.** Concentration-dependent binding data for PKIVIT flank library.
DOI: https://doi.org/10.7554/eLife.40499.042

**Source data 5.** Concentration-dependent binding data for NFATc2 core library.
DOI: https://doi.org/10.7554/eLife.40499.043

**Source data 6.** Concentration-dependent binding data for NFATc2 flank library.
DOI: https://doi.org/10.7554/eLife.40499.044

**Source data 7.** Concentration-dependent binding data for AKAP79 core library.
DOI: https://doi.org/10.7554/eLife.40499.045

**Source data 8.** Concentration-dependent binding data for AKAP79 flank library.
DOI: https://doi.org/10.7554/eLife.40499.046

**Source data 9.** WT and mutant binding data for PVIVIT core library.
DOI: https://doi.org/10.7554/eLife.40499.047

**Source data 10.** WT and mutant binding data for PVIVIT, PKIVIT, NFACTc2, and AKAP79 core and flank libraries.
DOI: https://doi.org/10.7554/eLife.40499.048

**Figure supplement 1.** Concentration-dependent binding measurements for systematic mutations within the PVIVIT core motif.
DOI: https://doi.org/10.7554/eLife.40499.016

**Figure supplement 2.** Concentration-dependent binding measurements for systematic mutations flanking the PVIVIT motif.
DOI: https://doi.org/10.7554/eLife.40499.017

**Figure supplement 3.** Concentration-dependent binding measurements for systematic mutations within the PKIVIT core motif.
DOI: https://doi.org/10.7554/eLife.40499.018

**Figure supplement 4.** Concentration-dependent binding measurements for systematic mutations flanking the PKIVIT motif.
DOI: https://doi.org/10.7554/eLife.40499.019

**Figure supplement 5.** Concentration-dependent binding measurements for systematic mutations within the NFATc2 core motif.
DOI: https://doi.org/10.7554/eLife.40499.020

**Figure supplement 6.** Concentration-dependent binding measurements for systematic mutations flanking the NFATc2 motif.
DOI: https://doi.org/10.7554/eLife.40499.021

**Figure supplement 7.** Concentration-dependent binding measurements for systematic mutations within the AKAP79 core motif.
DOI: https://doi.org/10.7554/eLife.40499.022

**Figure supplement 8.** Concentration-dependent binding measurements for systematic mutations flanking the AKAP79 motif.
DOI: https://doi.org/10.7554/eLife.40499.023

**Figure supplement 9.** Binding intensity comparisons for PVIVIT library.
DOI: https://doi.org/10.7554/eLife.40499.024

**Figure supplement 10.** Binding intensity comparisons for PKIVIT library.
DOI: https://doi.org/10.7554/eLife.40499.025

**Figure supplement 11.** Binding intensity comparisons for NFATc2 library.
DOI: https://doi.org/10.7554/eLife.40499.026

**Figure supplement 12.** Binding intensity comparisons for AKAP79 library.
DOI: https://doi.org/10.7554/eLife.40499.027

**Figure supplement 13.** Bar graphs showing PVIVIT specificity.
DOI: https://doi.org/10.7554/eLife.40499.028

**Figure supplement 14.** $\Delta\Delta G$ calculated in reference to PxIxIT WT (HPVIVITGPH).
DOI: https://doi.org/10.7554/eLife.40499.029

**Figure supplement 15.** Relative affinity was calculated by normalizing all binding affinity ($K_d$) to WT binding affinity (light blue: core, dark blue: flank).
DOI: https://doi.org/10.7554/eLife.40499.030

**Figure supplement 16.** $\Delta\Delta G$ calculated in reference to PKIVIT WT.
DOI: https://doi.org/10.7554/eLife.40499.031

*Figure 3 continued*

**Figure supplement 17.** Relative affinity was calculated by normalizing all binding affinity ($K_d$) to WT (SPRIEITPSH) binding affinity (light blue: core, dark blue: flank).
DOI: https://doi.org/10.7554/eLife.40499.032

**Figure supplement 18.** $\Delta\Delta G$ calculated in reference to WT peptide (SPRIEITPSH).
DOI: https://doi.org/10.7554/eLife.40499.033

**Figure supplement 19.** Relative affinity was calculated by normalizing to peptides KRMEPIAIIITDTEIS.
DOI: https://doi.org/10.7554/eLife.40499.034

**Figure supplement 20.** $\Delta\Delta G$ calculated in reference to peptide KRMEPIAIIITDTEIS.
DOI: https://doi.org/10.7554/eLife.40499.035

**Figure supplement 21.** Visualization of Log $K_a$ normalized to corresponding WT sequences.
DOI: https://doi.org/10.7554/eLife.40499.036

**Figure supplement 22.** LogoMaker logos for binding data.
DOI: https://doi.org/10.7554/eLife.40499.037

**Figure supplement 23.** Measured changes in binding affinity for mutations in PVIVIT (top panel) and PKIVIT (bottom panel) relative to the same mutations in the NFATc2 (PRIEIT) library.
DOI: https://doi.org/10.7554/eLife.40499.038

*22*) (*Tareen and Kinney, 2019*). Although considered to be degenerate in the consensus motif, many substitutions at positions 2 and 4 had unexpectedly strong effects on binding (*Figure 3B*). In particular, V2R and V2K mutations in the PVIVIT scaffold significantly enhanced binding in experimental assays, consistent with previous observations of high-affinity CN binding to the viral inhibitor (A238L) containing a positively-charged residue at this position (PKIIIT) (*Grigoriu et al., 2013*).

## Flanking residues play a major role in defining binding affinity and specificity

The experimental impact of sequences flanking the core 'PxIxIT' residues on CN binding has never been probed systematically. At the −1 position, MRBLE-pep data revealed a clear CN preference for hydrophobic residues (V,L,I) and a weak preference for polar residues (T, Y, H, N, and R) for all three of the PVIVIT, PKIVIT, and NFATc2 scaffolds. By contrast, substitution to acidic (D, E) residues at this position strongly reduced binding (*Figure 3C*, *Figure 3—figure supplement 1*, *Figure 3—figure supplement 2*, *Figure 3—figure supplement 3*, *Figure 3—figure supplement 4*, *Figure 3—figure supplement 5*, *Figure 3—figure supplement 6*, *Figure 3—figure supplement 13*, *Figure 3—figure supplement 14*, *Figure 3—figure supplement 15*, *Figure 3—figure supplement 16*, *Figure 3—figure supplement 17*, *Figure 3—figure supplement 18*). At position 9, the majority of mutations in the PVIVIT scaffold increased affinity. Substitution of phosphomimetic residues (D, E) or unphosphorylated serine increased affinity slightly and not at all, respectively, but substitution of phosphoserine and phosphothreonine residues (X, Z) increased affinity nearly 50-fold (from $K_d$ = 170 + /- 40 nM to $K_d$ = 3.4 + /- 0.2 nM) (*Figure 3C–D*, *Figure 3—figure supplement 2*). This effect was specific to the PVIVIT sequence, with significantly less drastic increases and decreases in affinity observed for the same substitutions in the NFATc2 and AKAP79 scaffolds, respectively (*Figure 3—figure supplement 6*, *Figure 3—figure supplement 8*, *Figure 3—figure supplement 23*). These MRBLE-pep data establish that flanking residues make major contributions to affinity and the PxIxIT SLiM is significantly longer than previously thought.

## Effects of individual substitutions show evidence of non-additivity

Understanding the degree to which CN-SLiM binding specificity can be explained by a linear additive model is critical for estimating the likely accuracy of downstream efforts to identify substrates or design therapeutics. Deviations from additivity can be quantified via double-mutant cycle (DMC) analysis, in which the effects of individual mutations are measured alone and in combination and then compared. DMC analysis between the PVIVIT and PKIVIT scaffolds revealed that the V4F substitution resulted in a significant loss of affinity within the PVIVIT context but had only minor effects on PKIVIT binding; by contrast, V4I mutation of this solvent-exposed position significantly increased binding only within the PKIVIT sequence (*Figure 3E*, *Figure 3—figure supplement 23*). This last substitution suggests that the high affinity of the PKIIIT sequence in the A238L inhibitor in particular

relies on cooperativity between the position 2 and 4 residues. The magnitude of $\Delta\Delta G$ effects for the same substitution at the same position varies significantly between different backbones, further illustrating the importance of epistatic interactions in dictating overall binding strength. Although critical for improved prediction of *in vivo* substrates and outputs, these epistatic interactions are obscured in typical high-throughput screens and can only be observed in systematic pairwise measurements like those presented here.

## Comparing experimental measurements with results from structure-based computational predictions

To test the degree to which computational methods can quantitatively predict effects of amino acid substitutions within SLiMs on protein-SLiM binding affinities, we leveraged high-resolution co-crystal structures and recently-developed computational modeling techniques (*Smith and Kortemme, 2010*; *Barlow et al., 2018*; *Guerois et al., 2002*; *Schymkowitz et al., 2005*) to estimate the effects of each mutation. Co-crystal structures of CN bound to PVIVIT (PDB: 2P6B) (*Li et al., 2007*) and AKAP79-derived PIAIIIT (PDB: 3LL8) (*Li et al., 2012*) reveal that the conserved P1, I3, and I5 residues are buried in hydrophobic pockets on the CN surface in all three structures; for AKAP79, an I1 residue occupies the pocket typically occupied by a P1 residue. In both structures, the less-conserved solvent-exposed (positions 2 and 4) and flanking (positions −1 and 7–9) residues adopt variable side chain orientations (*Figure 4A*).

To predict mutational effects, we used computational methods implemented in the protein structure prediction and design programs Rosetta (*Leaver-Fay et al., 2011*) and FoldX (*Guerois et al., 2002*; *Schymkowitz et al., 2005*). The Rosetta Sequence Tolerance protocol (*Smith and Kortemme, 2010*) samples rotamers of different amino acid residues at each position and returns a qualitative estimate of effects of substitutions on binding (*Figure 4—figure supplement 2*, *Figure 4—source data 1*, *Figure 4—source data 2*). The Rosetta 'flex_ddG' protocol (*Barlow et al., 2018*) quantitatively estimates effects of mutations on binding free energies for systematic mutations at each position (*Figure 4B*, *Figure 4—source data 3*, *Figure 4—source data 4*), and has been shown to perform comparably to more computationally expensive alchemical free energy calculations for predicting mutational effects on protein-ligand affinities (*Aldeghi et al., 2018*). As expected, substitutions at invariant positions P1 and I3 were generally predicted to be destabilizing, while changes to the solvent-exposed, degenerate x2 and x4 positions were predicted to have minimal effects (*Figure 4B*). Intriguingly, mutations within upstream and downstream residues were frequently predicted to enhance binding. Mutations to more hydrophobic or aromatic residues (W,F,Y, I,V) in the −1 position for the PVIVIT motif (*Figure 4B*, left) or to most residues other than the native P-1 for the AKAP79 motif (*Figure 4B*, right) improved binding in silico, as did mutations at Position eight to aromatic (F,Y), aliphatic (I, V, L), or acidic (D, E) residues (*Figure 4B*). The FoldX protocol uses an empirical force field to allow rapid evaluation of the effects of mutations on the free energy of protein folding and complex formation (*Guerois et al., 2002*; *Schymkowitz et al., 2005*). These calculations revealed globally similar results (*Figure 4C*, *Figure 4—figure supplement 2*, *Figure 4—figure supplement 3*, *Figure 4—source data 5*, *Figure 4—source data 6*), with mutations to conserved hydrophobic residues again predicted to be destabilizing and mutations to other residues predicted to have little effect. However, Rosetta predicted less deleterious effects for substituting glycine and proline residues, perhaps due to better local sampling of backbone conformations.

To determine the degree to which Rosetta and FoldX successfully predict observed mutational effects on binding, we generated scatter plots showing relationships between measured affinities and predicted frequencies or changes in free energy of binding ($\Delta\Delta G$), respectively, for each residue at each position (*Figure 4—figure supplement 4*; *Figure 4—figure supplement 5*, *Figure 4—figure supplement 6*). All methods correctly predicted that I3 and I5 mutations decrease affinity; however, modeling approaches predicted that variations at solvent-exposed positions (V2 and V4) would have little effect on affinity, in direct contrast to experimental observations (*Figure 4—figure supplement 4*, *Figure 4—figure supplement 5*, *Figure 4—figure supplement 6*). Calculations of the fraction of residues correctly predicted to be either stabilizing ($\Delta\Delta G < 0$) or destabilizing ($\Delta\Delta G > 0$) by each algorithm at each position within the PVIVIT and AKAP79 peptides confirm relatively strong performance for buried hydrophobic residues with variable performance at other positions (*Figure 4D*), with FoldX generally slightly outperforming Rosetta flex_ddG. These results highlight difficulties associated with properly modeling conformational changes in solvent-exposed and non-

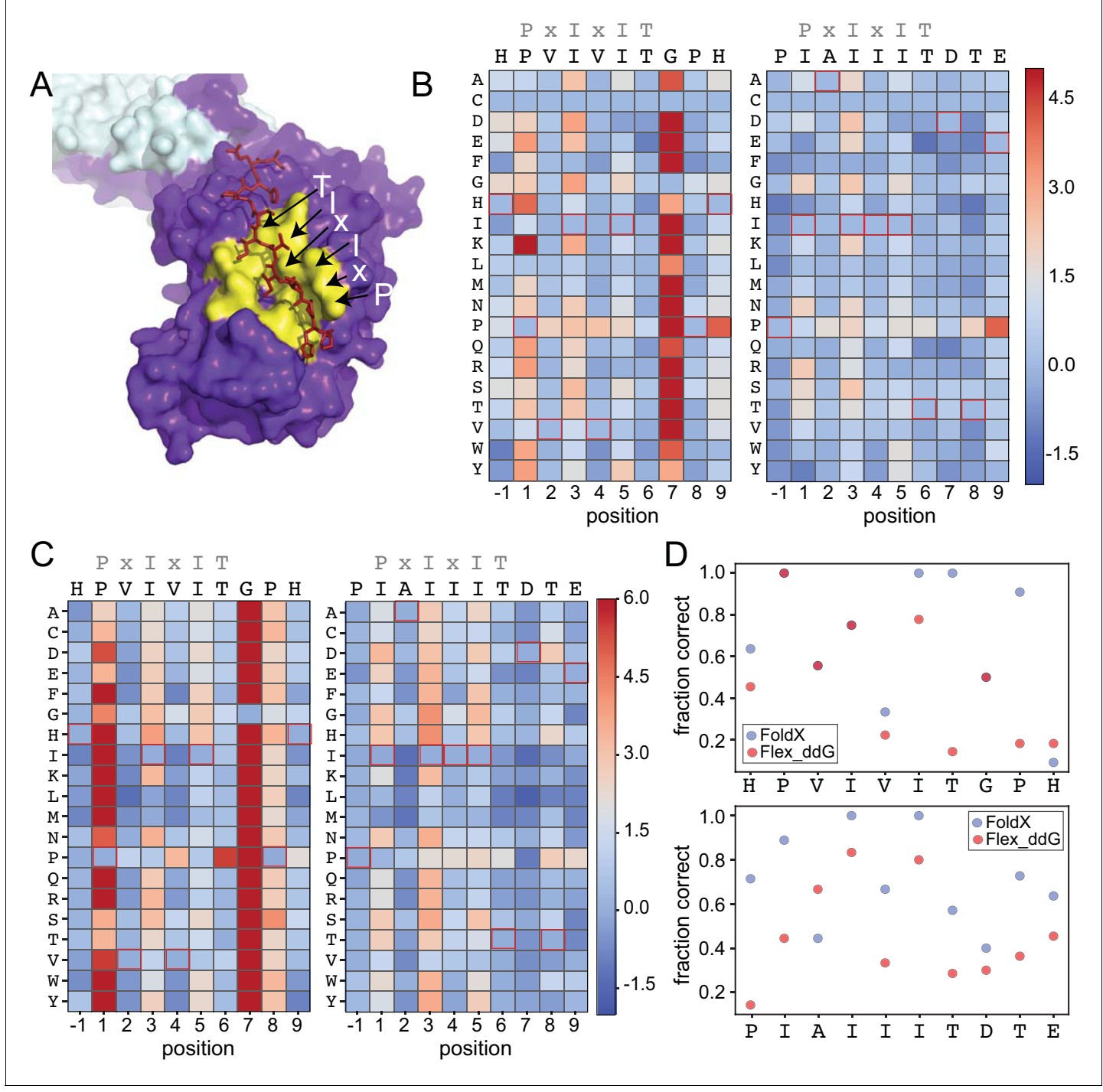

**Figure 4.** Structure-based modeling predictions. (**A**) Co-crystal structures showing zoomed in overlay of substrates (PDB: 2P6B). (**B**) Heat maps showing predicted changes in binding energy from the Rosetta 'flex_ddG' protocol for each amino acid substitution at each position for PVIVIT and IAIIIT targets. (**C**) Heat maps showing FoldX-predicted changes in binding energy for each amino acid substitution at each position for PVIVIT and IAIIIT targets. (**D**) Fraction of residues correctly predicted to be stabilizing or destabilizing by both models as a function of position for PVIVIT (top) and IAIIIT (bottom).

DOI: https://doi.org/10.7554/eLife.40499.049

The following source data and figure supplements are available for figure 4:

**Source data 1.** Rosetta sequence tolerance protocol frequencies for PVIVIT.
DOI: https://doi.org/10.7554/eLife.40499.057

**Source data 2.** Rosetta sequence tolerance protocol frequencies for IAIIIT.

*Figure 4 continued on next page*

*Figure 4 continued*

DOI: https://doi.org/10.7554/eLife.40499.058

**Source data 3.** Flex ddG-predicted $\Delta\Delta G$ values for PVIVIT.

DOI: https://doi.org/10.7554/eLife.40499.059

**Source data 4.** Flex ddG-predicted $\Delta\Delta G$ values for IAIIIT.

DOI: https://doi.org/10.7554/eLife.40499.060

**Source data 5.** FoldX ddG-predicted $\Delta\Delta G$ values for PVIVIT.

DOI: https://doi.org/10.7554/eLife.40499.061

**Source data 6.** FoldX ddG-predicted $\Delta\Delta G$ values for IAIIIT.

DOI: https://doi.org/10.7554/eLife.40499.062

**Figure supplement 1.** Rosetta Sequence Tolerance protocol frequencies.

DOI: https://doi.org/10.7554/eLife.40499.050

**Figure supplement 2.** FoldX versus Rosetta predictions for PVIVIT.

DOI: https://doi.org/10.7554/eLife.40499.051

**Figure supplement 3.** FoldX versus Rosetta predictions for IAIIIT.

DOI: https://doi.org/10.7554/eLife.40499.052

**Figure supplement 4.** Measured affinities versus predicted amino acid frequencies.

DOI: https://doi.org/10.7554/eLife.40499.053

**Figure supplement 5.** Measured versus Rosetta-predicted changes in Gibbs free energies.

DOI: https://doi.org/10.7554/eLife.40499.054

**Figure supplement 6.** Measured versus FoldX-predicted changes in Gibbs free energies.

DOI: https://doi.org/10.7554/eLife.40499.055

**Figure supplement 7.** MRBLE-pep measurements, FoldX predictions, and Rosetta predictions versus orthogonal affinity measurements for 3 PVIVIT variants.

DOI: https://doi.org/10.7554/eLife.40499.056

conserved positions, and show the utility of quantitative experimental measurements such as those presented here for improving binding models. Thermodynamic Integration and Free Energy Perturbation (TI/FEP) methods would likely enhance the ability to predict effects of mutating solvent-exposed residues, but are significantly more computationally expensive (*Kilburg and Gallicchio, 2016*; *Gallicchio and Levy, 2011*). Comparisons of MRBLE-pep measurements and Rosetta and FoldX predictions of the energetic effects of single site substitutions with orthogonal measurements are complicated by a lack of previously published affinity data. However, a direct comparison between MRBLE-pep measurements and computational predictions for two single-site substitutions in the PVIVIT motif reveal that only MRBLE-pep correctly discerns that a PVIVVT substitution has little effect while a PVIVIN substitution reduces binding (*Figure 4—figure supplement 7*).

## Absolute binding affinities confirm that mutations to flanking residues dramatically change affinities

To determine absolute affinities for peptides spanning a range of binding behaviors, we performed two experiments measuring concentration-dependent CN binding to an additional 'calibration' library containing ~6 peptides from each core and flank library (*Figure 5—figure supplement 1*, *Figure 5—figure supplement 2*, *Figure 5—source data 1*, *Figure 5—source data 2*). These experiments represented full technical replicates in which peptides were synthesized de novo on MRBLEs produced at different times and assayed for binding using different batches of purified calcineurin. Measured $\Delta\Delta G$ values relative to the PVIVIT variant between experiments showed strong agreement (Pearson $r^2$ = 0.72 (*Figure 5—figure supplement 3*), demonstrating the robustness of the assay. Averaged absolute affinities estimated for all variants using these $\Delta\Delta G$ values and the known literature $K_d$ for PVIVIT confirm that hydrophobic residues at position −1 and phosphorylated residues at position nine significantly increase affinity by 10-fold and 100-fold, respectively (*Figure 5—figure supplement 4*).

## Identification of high-affinity scaffolds with therapeutic potential

Although cyclosporin A and FK506 are routinely prescribed to transplant patients to inhibit CN-dependent immune response activation, both drugs are associated with adverse effects that likely

result from inhibition of CN-substrate dephosphorylation in non-immune tissues. While peptides themselves are rarely used as drugs, high-affinity peptides with increased specificity for a subset of CN-SLiM interactions could serve as initial scaffolds for downstream medicinal chemistry efforts to identify improved inhibitors. To probe for high-affinity variants, we measured CN binding to a final MRBLE-pep library containing the set of 36 'calibration' peptides along with 11 peptides containing combinations of mutations discovered to increase affinity (*Figure 5—figure supplement 5*, *Figure 5—figure supplement 6*, *Figure 5—figure supplement 7*, *Figure 5—source data 3*, *Figure 5—source data 4*). PVIVIT variants combining a phosphothreonine (Z) at position nine with hydrophobic residues (I, V) at position −1 showed the strongest binding, with measured $K_d$ values of ~10 nM (50x stronger than the measured PVIVIT affinity), and multiple PVIVIT, PKIVIT, and AKAP79 variants also showed enhanced binding (*Figure 5A*). Measured $\Delta\Delta G$s for these full technical replicates showed remarkable agreement ($r^2 = 0.83$) (*Figure 5B*), establishing the robustness and reproducibility of the assay.

## Quantitative comparisons between *in vitro* affinities and *in vivo* inhibition via calcineurin activity assays

The ultimate goal of mapping CN-SLiM binding energy landscapes is to improve understanding of CN target substrate recognition and enable rational design of *in vivo* inhibitors. To test the degree to which *in vitro* MRBLE-pep affinities predict *in vivo* inhibition, we used a previously developed dual luciferase reporter assay to assess the activity of the NFAT2 transcription factor in HEK293T cells (*Figure 6A*) (*Grigoriu et al., 2013*). NFAT2 must be dephosphorylated by CN to accumulate in the nucleus, where it activates transcription of its target genes, and this interaction can be inhibited by peptides or small molecules that binds to CN. Reduction of NFAT2-dependent transcriptional activity upon expression of a competing PxIxIT peptide therefore reflects, quantitatively, the affinity of the inhibitor peptide for CN. Candidate PxIxIT inhibitors were co-expressed as GFP-fusion proteins within cells to enhance their stability and **facilitate** direct measurement of inhibitor concentrations.

We compared levels of NFAT-driven luciferase activity in the presence of empty vector, vectors driving expression of the 4 PxIxIT peptide scaffolds probed extensively *in vitro* (PVIVIT, PKIVIT, NFATc2, and AKAP79), or FK506, a small-molecule inhibitor of CN with a previously measured IC50 of ~0.5 nM ( *Figure 6B*, *Figure 6—source data 1*, *Figure 6—source data 2*, *Figure 6—source data 3*) (*Fruman et al., 1992*). All 4 PxIxIT peptides substantially inhibited NFAT-driven luciferase activity, confirming competitive inhibition of CN-NFAT-(PxIxIT) binding *in vivo*. Consistent with MRBLE-pep measurements, PKIVIT was the most effective inhibitor, followed closely by PVIVIT. NFATc2 and AKAP79 showed weaker *in vivo* inhibition, consistent with MRBLE-pep results and in contrast with prior work suggesting that AKAP79 binds with equal affinity to PVIVIT ($K_d$~500 nM) (*Li et al., 2012*). Co-expression of the PKIVIT peptide resulted in greater inhibition than FK506, which interferes with CN-substrate interactions by occluding the LxVP-SLiM binding site (*Rodríguez et al., 2009*) and is a potent CN inhibitor.

We next examined the importance of PxIxIT flanking residues by creating substitutions in the −1 and 9 positions (*Figure 6C,D*). Consistent with in silico predictions and MRBLE-pep affinities, substitution of a hydrophobic residue (IPKIVIT or LPRIEIT) at the −1 position significantly increased inhibition, whereas incorporating an acidic residue (EPKIVIT or DPRIEIT) greatly diminished inhibition. As predicted computationally and observed in the MRBLE-pep assays, introduction of the phosphomimetic mutation H9D resulted in slightly increased inhibition; it was not possible to directly test the extremely high affinity phosphoserine or phosphothreonine *in vivo*. Taken together, in silico modeling, the MRBLE-pep affinities, and *in vivo* results suggest that phosphorylation at position nine may be a key determinant of modulating PxIxIT-CN affinity.

Finally, we sought to test whether MRBLE-pep *in vitro* affinity measurements could identify initial peptide scaffolds exhibiting strong inhibition *in vivo* for future optimization as peptide or small-molecule inhibitors. PKIVIT was a significantly stronger *in vivo* inhibitor than the previously patented PVIVIT peptide (*Nakamura et al., 2007*), with activity comparable to the PKIIIT site within the known A238L inhibitor. As predicted, both the IPKIVITGPH and HPKIVITGPD variants exhibited even stronger binding and inhibition (*Figure 6D*); similarly, mutating the I1 and A2 residues in the AKAP79 PIAIIIT motif to create a novel PxIxIT sites with adjacent prolines and a positively charged residue in the second position (PPKIIIT) greatly enhanced inhibition. Overall, MRBLE-pep *in vitro* measurements were strongly predictive of *in vivo* inhibition (*Figure 6E*).

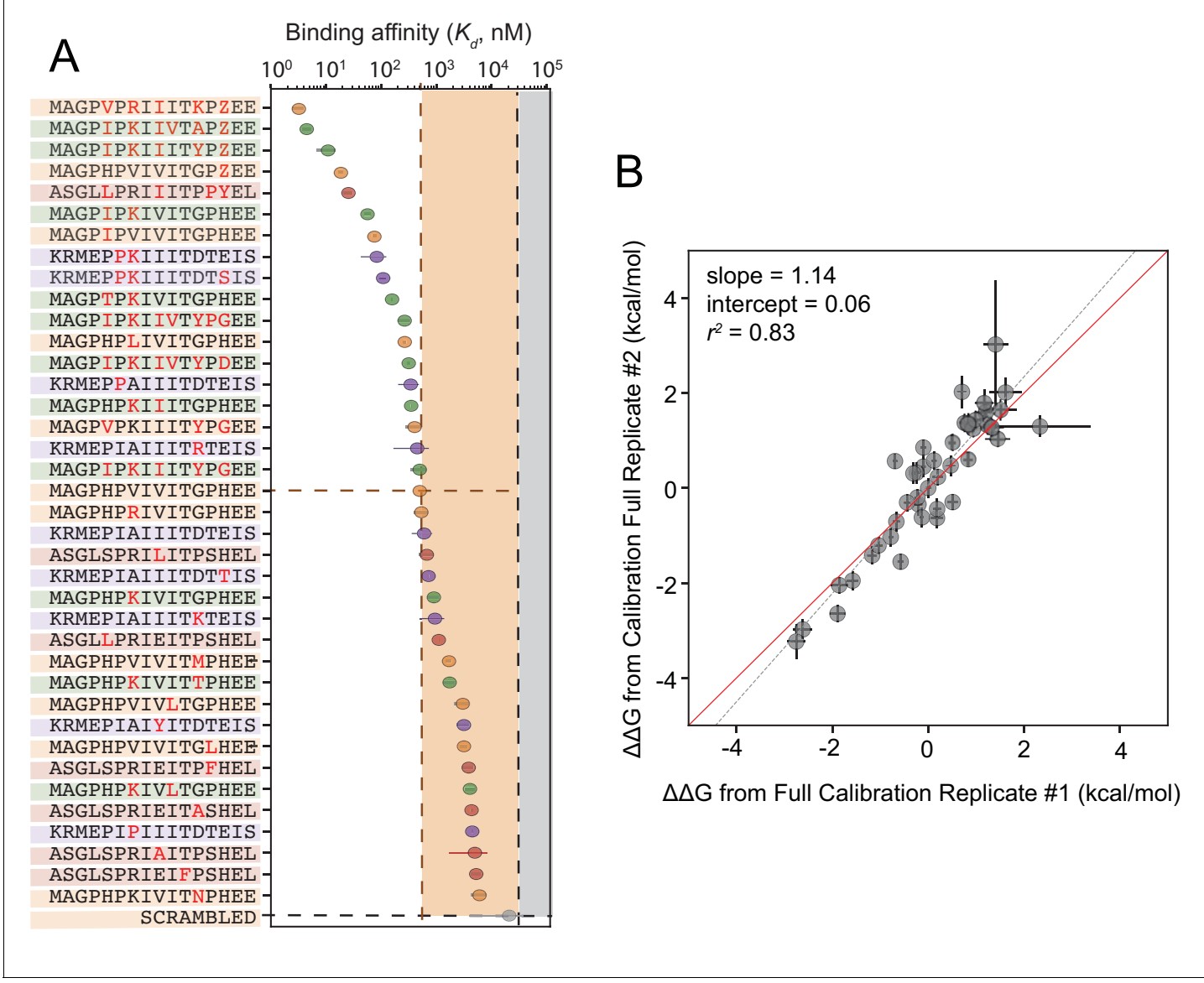

**Figure 5.** Calibrated *in vitro* affinities for a range of peptides. (**A**) *In vitro* measurements of binding affinities for high-affinity binders identified by combining favorable residues. (**B**) Correlation between measured ΔΔG values for a full technical replicate (new bead synthesis, new peptide synthesis, and new calcineurin protein purification) for the library shown in (**A**). Solid red line shows expected 1:1 linear relationship, dotted grey line shows linear regression of log-transformed values.

DOI: https://doi.org/10.7554/eLife.40499.063

The following source data and figure supplements are available for figure 5:

**Source data 1.** Concentration-dependent binding for calibration library, replicate 1.
DOI: https://doi.org/10.7554/eLife.40499.071

**Source data 2.** Concentration-dependent binding for calibration library, replicate 2.
DOI: https://doi.org/10.7554/eLife.40499.072

**Source data 3.** Concentration-dependent binding for full calibration library, replicate 1.
DOI: https://doi.org/10.7554/eLife.40499.073

**Source data 4.** Concentration-dependent binding for full calibration library, replicate 2.
DOI: https://doi.org/10.7554/eLife.40499.074

**Figure supplement 1.** Concentration-dependent binding for calibration peptide library, replicate 1.
DOI: https://doi.org/10.7554/eLife.40499.064

**Figure supplement 2.** Concentration-dependent binding for calibration peptide library, replicate 2.
DOI: https://doi.org/10.7554/eLife.40499.065

*Figure 5 continued on next page*

*Figure 5 continued*

**Figure supplement 3.** Correlation between $\Delta\Delta G$ measurements for calibration library technical replicates.
DOI: https://doi.org/10.7554/eLife.40499.066

**Figure supplement 4.** $K_d$ values for all peptides within calibration library.
DOI: https://doi.org/10.7554/eLife.40499.067

**Figure supplement 5.** Concentration-dependent binding for full calibration library, replicate 1.
DOI: https://doi.org/10.7554/eLife.40499.068

**Figure supplement 6.** Concentration-dependent binding for full calibration library, replicate 2.
DOI: https://doi.org/10.7554/eLife.40499.069

**Figure supplement 7.** Correlation between $\Delta\Delta G$ measurements for core/flank library measurements and 'calibration' library measurements.
DOI: https://doi.org/10.7554/eLife.40499.070

## Discussion

Here, we demonstrate a novel strategy to quantitatively profile parts of the binding specificity landscape for a peptide-protein interaction, thereby generating key insights required to map and model essential signaling networks in healthy and diseased cells. Our approach is complementary to current techniques, with the potential to provide new insights into SLiM recognition and function throughout signaling networks. Yeast and phage display can powerfully discriminate between 'bound' and 'unbound' peptide populations (*Bazan et al., 2012*; *Cherf and Cochran, 2015*) for > 10$^8$ protein-peptide interactions, and array-based methods can return information about qualitative binding strengths for many protein-peptide interactions in parallel. However, neither method can return the quantitative thermodynamic constants (dissociation constants ($K_d$s) and relative binding energies ($\Delta\Delta G$s) essential for predicting protein occupancies *in vivo* and effects on downstream signaling responses. For display- and array-based screens, measurements take place far from equilibrium, biasing recovered interactions towards the strongest binding motifs and losing information about those interactions with $K_d$s close to physiological concentrations most likely to be dynamically regulated *in vivo*.

The MRBLE-pep assay is faster and requires less in material than alternative low-throughput quantitative methods: measuring interaction affinities for 384 peptides using MRBLE-pep requires ~400x, ~700x, and ~2000x less purified protein than surface plasmon resonance, fluorescence polarization, and isothermal calorimetry, respectively, with savings increasing with library size (*Supplementary file 2*). This reduction in material should allow future profiling of a wide range of biologically important SLiM-binding domains, including those that are uncharacterized, unstable, and/or difficult to express recombinantly. MRBLE-pep libraries can be synthesized, assayed, and imaged in days, facilitating iterative rounds of synthesis and measurement for efficient affinity landscape mapping as well as testing and refining computational prediction algorithms. After screening large sequence spaces for candidate interaction motifs, MRBLE-pep can be used to quickly and efficiently scan through thousands of potential SLiMs to validate true hits, measure affinities for these hits in high-throughput, and systematically perturb amino acids within motifs to derive a position-specific affinity matrix that can be used to predict additional candidate downstream targets *in vivo*. Moreover, the ability to tune the density of applied peptides enables fast optimization to probe the affinity range of interest as well as test for surface density-induced binding artifacts. An elegant 'hold-up' assay that couples high-throughput affinity purification to microfluidic gel electrophoresis recently demonstrated a similar ability to measure relative affinities for 400 domain-peptide pairs (*Vincentelli et al., 2015*). However, this method requires considerably more infrastructure than the MRBLE-pep assay and has not yet been used in subsequent work. In future work, the MRBLE-pep platform could be combined with a simple and portable imaging device to allow labs to employ the assay with only standard laboratory equipment (*e.g.* pipettes and a benchtop centrifuge).

MRBLE-pep measurements can also be combined with data from high-throughput *in vivo* methods to decipher the structure and wiring of cellular signaling networks. Yeast two-hybrid (Y2H) assays and affinity purification coupled to mass spectrometry (AP-MS) techniques have revealed tens of thousands of new interactions within the human interactome (*Rolland et al., 2014*; *Huttlin et al., 2015*; *Huttlin et al., 2017*). However, SLiM-mediated binding interactions remain statistically underrepresented in these datasets (*Neduva and Russell, 2006*; *Van Roey et al., 2014*; *Davey et al.,*

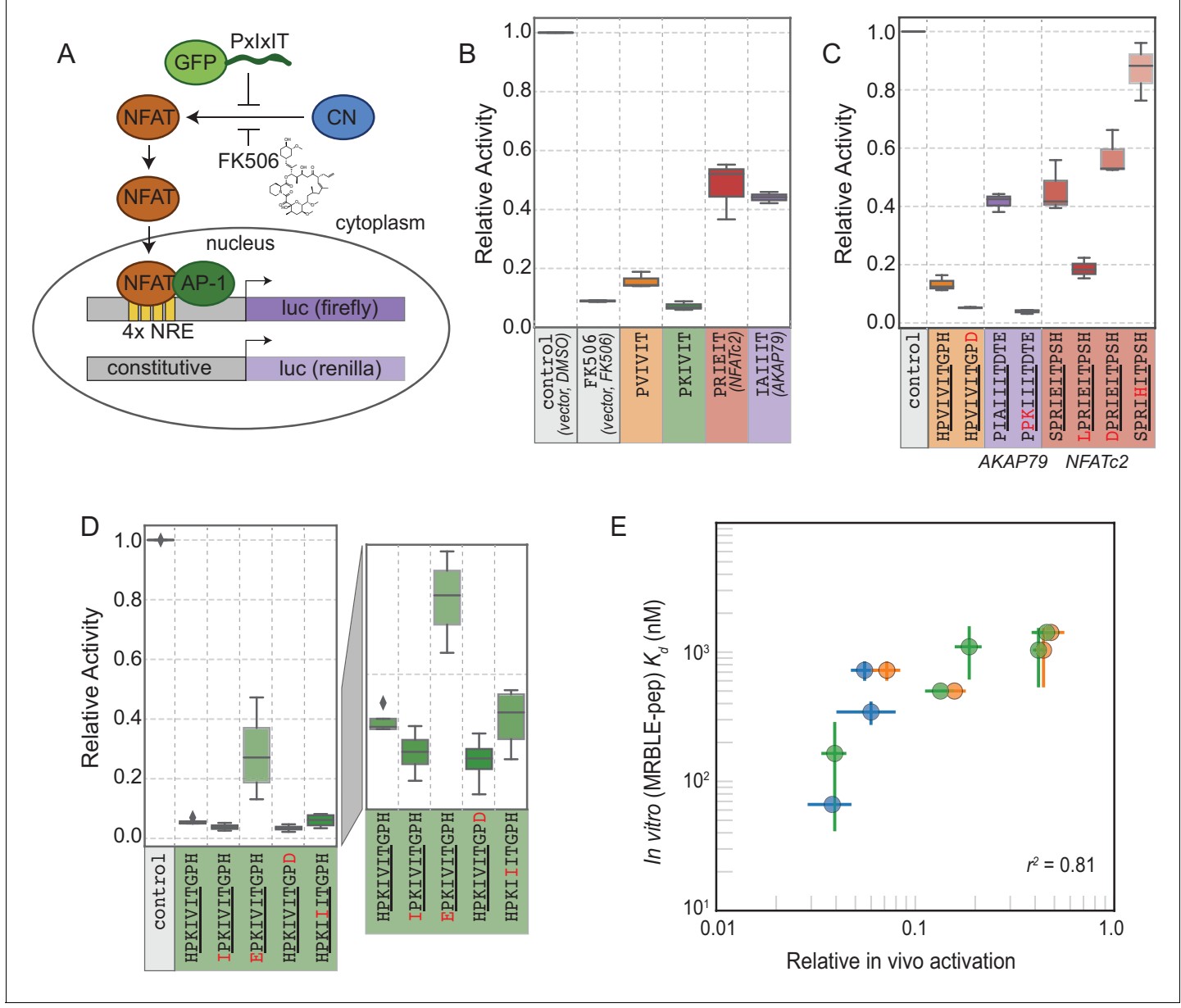

**Figure 6.** *In vivo* assays to quantify CN inhibition and an *in vitro* search for a potent inhibitor. (A) *In vivo* experimental assay schematic. (B) Reporter gene inhibition in the presence of an empty eGFP vector (control), an empty eGFP vector with topical application of FK506, and vectors expressing PVIVIT, PKIVIT, NFATc2, and AKAP79 PxIxIT peptides with a C-terminal eGFP tag. (C) Reporter gene inhibition in the presence of various PVIVIT-, AKAP79-, and NFATc2-eGFP variants. (D) Reporter gene inhibition in the presence of various PKIVIT-eGFP variants. (E) *In vitro* calibrated binding affinities ($K_d$, nM) plotted against relative *in vivo* activation for data shown in panels B (orange), C (green), and D (blue).

DOI: https://doi.org/10.7554/eLife.40499.075

The following source data is available for figure 6:

**Source data 1.** Initial *in vivo* experiment.
DOI: https://doi.org/10.7554/eLife.40499.076
**Source data 2.** PKIVIT *in vivo* experiment.
DOI: https://doi.org/10.7554/eLife.40499.077
**Source data 3.** All other peptides *in vivo* experiment.
DOI: https://doi.org/10.7554/eLife.40499.078

*2012*), likely due to fast dissociation rates, and AP-MS results cannot distinguish between binary interactions and higher-order complexes, and provide no direct information about the motif involved. MRBLE-pep measurements of peptides tiled across candidate interacting proteins could confirm candidate interactors, identify the SLiMs responsible, and determine whether a candidate interaction is direct or indirect. In particular, the ability to return information about residues that ablate binding can significantly improve the ability to predict true targets *in vivo* (*Krystkowiak et al., 2018*). To test candidate inhibitor specificities, MRBLE-pep could profile binding of a broad set of cellular proteins against high-affinitiy inhibitors with unnatural amino acids returned from recently developed high-throughput mass spectrometry-based (LC/MS/MS) approaches (*Gates et al., 2018*; *Vinogradov et al., 2017*).

Beyond the method itself, our findings reveal biologically significant insights into CN-PxIxIT specificity and illustrate how systematic analyses can shed light on elusive SLiM-protein interactions. The MRBLE-pep data and *in vivo* validation presented here are the first to reveal the impact of PxIxIT flanking residues on affinity, allowing identification of several peptides whose affinity for CN is two orders of magnitude higher than that of the highest affinity known substrate (PVIVIT) and potently inhibit CN-NFAT signaling *in vivo*. We reveal the presence of significant epistasis between residues, including one interaction important for the high affinity of a known CN inhibitor (A238L) (*Figure 3E*). Finally, we provide affinity measurements for a collection of peptides based on different known PxIxIT scaffolds with a wide variety of affinities, yielding a tool kit of peptide sequences that can be used to probe CN recognition and function *in vivo*.

In addition to identifying candidate peptides for manipulation of CN signaling, this first demonstration that residues both upstream and downstream contribute to PxIxIT-CN affinity redefines the motif itself. Current computational strategies attempt to identify novel CN substrates by searching for sequences within the proteome that match a consensus for the six core residues alone. The expanded 10-residue definition of the PxIxIT motif derived here will significantly enhance future computational efforts to comprehensively map the CN signaling network, reducing the number of both false positive and false negative substrates. In addition, MRBLE-pep provides critical Information about residues that reduce affinity (*e.g.* acidic residues at the −1 position reduces affinity) that is never revealed by positive screening methods and is rarely collected due to the resource-intensive nature of such analyses using established techniques. The ability to predict affinity based on SLiM sequence is an essential step toward accurate modeling of signaling dynamics *in vivo*, as the extent of *in vivo* dephosphorylation has previously been shown to depend on primary PxIxIT amino acid sequence (and the associated interaction affinity) (*Roy et al., 2007*). Even subtle (~2-fold) differences in binding could have profound effects on downstream signaling under physiologic conditions given the low affinities of CN-PxIxIT interactions.

Establishing the impact of post translational modifications on SLiM-protein affinity is critical for understanding cross talk between signaling networks *in vivo*. Consistent with our findings that acidic residues at the −1 position decrease affinity, JNK kinase regulates CN signaling by phosphorylating a serine that immediately precedes the PxIxIT of NFAT4 (*Chow et al., 2000*). Similarly, our demonstration that phosphorylated residues downstream of the core PxIxIT sequence at position nine enhance PVIVIT affinity echoes observations that phosphorylation of a threonine in this position is required for binding of CN to a PxIxIT site in C16ORF74, and for its ability to promote invasiveness of pancreatic ductal adenocarcinoma (PDAC) (*Nakamura et al., 2017*). This positive effect on binding is context-dependent, increasing PVIVIT binding 50-fold but having little or no effect on other PxIxIT sequences, reinforcing the importance of systematic analyses for generating predictive information.

Beyond improving the ability to reconstruct downstream CN signaling networks *in vivo*, these results also have direct relevance to precision medicine. Current annotations of 2RQ, 7 PA, and 7PL missense mutations within the NFATc2 PxIxIT site recovered from sequenced lung and breast adenocarcinomas describe them as unlikely to have functional effects *in vivo* (*cBioPortal, 2018*). While the experiments presented here did not directly test the functional consequences of a R2Q mutation, all mutations away from the native residue at this position resulted in a dramatic loss of binding (*Figure 3—figure supplement 17*), and the P7A substitution led to a nearly complete loss of binding (*Figure 3—figure supplement 6*, *Figure 3—figure supplement 17*, *Figure 3—figure supplement 18*). Thus, binding specificity maps like those obtained here could both help clinicians identify functionally significant missense mutations, and refine the computational algorithms that predict the

impact of such alleles. Overall, the approaches outlined here will help decipher the language of SLiM-domain binding events, and of the cellular signaling networks they define in both healthy and diseased cells.

## Materials and methods

Reagents for peptide synthesis were purchased and used without further purification from NovaBiochem, AnaSpec (Fremont, CA), and Sigma-Aldrich (St. Louis, MO). All other solvents and chemical reagents were purchased from Sigma-Aldrich.

### MRBLE synthesis and collection

MRBLEs were synthesized using a previously published microfluidic device (Gerver et al., 2012; Nguyen et al., 2017a; Nguyen et al., 2017b) with each code collected into wells of a 96-well plate using an open-source in-house fraction collector. Briefly, all lanthanide input mixtures contained double-distilled water (ddH$_2$O), 42.8% v/v 700 MW PEG-diacrylate (PEG-DA) (Sigma-Aldrich), 19.6 mg mL$^{-1}$ lithium phenyl-2,4,6-trimethylbenzoylphosphinate (LAP), and 5.0% v/v YVO4:Eu (at 25 mg mL$^{-1}$). Individual input mixtures contained 16.3% v/v of either YVO4:Sm (25 mg mL$^{-1}$), YVO4:Dy (25 mg mL$^{-1}$), or YVO4:Tm (12.5 mg mL$^{-1}$). Lanthanide solutions were mixed using a herringbone mixer channel and forced into droplets using a T-junction with a continuous stream of HFE7500 (3M Novec) containing 2% w/w modified ionic Krytox 157FSH (Miller Stephenson, Danbury, CT) (DeJournette et al., 2013). Droplets were then photopolymerized with UV light from a full-spectrum 200W Xenon arc lamp (Dymax, Torrington, CT, USA). With this setup, ~3000 beads containing each spectral code were synthesized in 70 s (as described previously Gerver et al., 2012; Nguyen et al., 2017a); for the 48-plex MRBLE library used here, each code was produced 10 times to yield ~30,000 beads per code.

### Bead functionalization

MRBLE hydrogel beads were functionalized with terminal amine handles via Michael addition by reacting available acrylates in the MRBLEs with a solution of cysteamine (50 eq) containing pyridine (50 eq) in H2O:DMF (1:3) for 18 hr at ambient temperature. Next, to selectively functionalize outer and inner regions, MRBLEs were swelled in water overnight and drained using a manifold before the addition of a solution containing Fmoc-N-hydroxysuccinimide (0.2 eq) and diisopropylethylamine (DIPEA) (0.8 eq) in dichloromethane:diethyl ether (55:45) with vigorous shaking (1600 rpm for 15 s followed by 30 s at rest) for 30 min (Liu et al., 2002). To selectively functionalize MRBLE inner core regions with an acid-labile rink amide linker, MRBLEs were then treated with 4-[(2,4-Dimethoxy-phenyl)(Fmoc-amino)methyl]phenoxyacetic acid (rink amide) (5.0 eq), N,N′-Diisopropylcarbodiimide (DIC) (5.0 eq), and DIPEA (10 eq) in dimethylformamide (DMF) for 1 hr and repeated twice. After removal of the Fmoc protecting group using 20% 4-methylpiperidine (4-MP) in DMF for 20 min, the effective on-bead peptide concentration was reduced by reacting the bead with a mixture of Fmoc-glycine-OH:Ac-N-glycine-OH (1:9) (5.0 eq), DIC(5.0 eq), and DIPEA (10 eq) for 14 hr. Following Fmoc deprotection, MRBLEs were transferred to an automated peptide synthesizer for solid phase peptide synthesis (SPPS).

### On-bead peptide synthesis

Peptide synthesis was performed using a Biotage Syro II automated peptide synthesizer following instructions from the manufacturer (Biotage, Charlotte, NC). During coupling steps, Fmoc-protected amino acids (10 eq) were activated with HCTU (9.8 eq) and NMM (20 eq) with coupling times of 8 min for standard amino acids and 25 min for phosphorylated amino acids. Each coupling round comprised of 2 sequential coupling reactions with the addition of fresh amino acid and coupling reagents. Deprotection was performed initially with 40% 4-MP for 2 min and then followed with 20% 4-MP for an additional 6 min. MRBLEs were then washed thoroughly with 6 rounds of DMF (0.4 mL) before the next coupling step. Before global deprotection with TFA, 50 μL from a volume of 400 μL for each code was saved for biotin conjugation (as described below). To perform the global deprotection step, MRBLEs were washed with DCM and dried under vacuum before the addition of 0.5 mL of TFA cocktail (Reagent B, TFA:phenol:ddH2O:triisopropylsilane, 88:5:5:2, v/m/v/v) was added to each reaction tube and reacted with shaking (15 s shaking and 1 min rest) for 1.5 hr. After TFA

deprotection, MRBLEs containing deprotected peptides were washed with TFA (~0.5 mL, and collected for MALDI analysis), DCM (~2 mL), neutralized with 10% DIPEA in DMF twice (~1 mL), and then finally washed with storage buffer (1 mL, 0.1% TBST with 0.02% NaN3) 3 times (*Kumaresan et al., 2011*). To assess peptide quality, the TFA solution containing cleaved/unprotected peptide was transferred to 15 mL falcon tubes using the provided liquid transfer system from Biotage. Peptides were triturated with cold diethylether (~1 mL), pelleted, decanted, and repeated these steps 3 times before preparing for MALDI analysis using a general protocol described below.

## Peptide concentration estimation via biotin conjugation assay

A small aliquot (~50 μL from 400 μL) of each code was transferred and pooled into a separate fritted reaction tube (Biotage 2 mL reaction tubes for SPPS) for biotin (50 eq) conjugation using DIC (50 eq), DIPEA (100 eq), and DMF overnight. After washing the MRBLEs (DMF, MeOH, and DCM), another round of coupling using the same conditions mentioned above was mixed for 2 hr and then washed. MRBLEs with terminal biotin were then globally deprotected using a cocktail of TFA/phenol/H2O/TIPS (87.5:5:5:2.5 v/v) at ambient temperature for 1.5 hr. After global deprotection, MRBLEs were neutralized with 10% DIPEA in DMF, washed with DCM, washed with 3 X PBS containing 0.1% TWEEN 20 (0.1% PBST), and stored in 0.1% PBST containing 0.02% NaN3. After biotin conjugation, a 40 μL aliquot from a 600 μL suspension of beads was passivated with 0.1% PBST containing 5% BSA in a 150 μL PCR strip tube on a rotator overnight at ~5 ˚C. The beads were then washed with 0.1% PBST containing 2% BSA 3 times (~100 μL), followed by incubation with 1 μL of labeled streptavidin (final concentration = ~189 nM, Abcam, ab134241) in 0.1% PBST containing 2% BSA (99 μL) for 30 min on a rotator at ambient temperature. After incubation, MRBLEs were pelleted and washed with 0.1% PBST once and then imaged to obtain binding data.

## MALDI quality control

After global deprotection, supernatants collected in 15 mL Falcon tubes for each code were placed into a freezer for 1 hr; these tubes were then centrifuged (4000 g for 20 min at 4 ˚C) and decanted (repeated 3 times). Peptides were then dissolved with 60% ACN/H2O (20 μL, phosphopeptides were dissolved with 50% acetic acid) for MALDI analysis using THAP (250 mM in ACN) as the matrix. To prepare the MALDI plate (microScout Target MSP 96 target polished steel BC, part #8280800), 0.5 μL of sodium citrate (250 mM in H2O containing 0.1% TFA) was spotted on to the plate surface and allowed to dry. After drying, 1 μL of a 1:1 mixture of the peptide solution with a solution of sodium citrate:THAP (1:1) was spotted onto the plate and allowed to dry again before analysis. Data was obtained using a Bruker microflex MALDI-TOF (Billerica, MA, USA). The instrument was run on positive-ion reflector mode with a laser setting of 1,810 V and data averaged over 100 scans. Raw data was analyzed using FlexAnalysis and mMass (ver. 5.5, http://www.mmass.org/).

## Purification of calcineurin

N-terminally, 6-His-tagged human calcineurin A isoforms (truncated at residue 400), either wild-type or containing the mutation 330NIR333-AAA, were expressed in tandem with the calcineurin B subunit in *E. coli* BL21 (DE3) cells (Invitrogen) and cultured in LB medium containing carbenicillin (50 mg/ml) at 37˚C to mid-log phase. Expression was induced with 1 mM isopropyl 1-thio-b-galactosidase at 16˚C for 18 hr. Cells were pelleted, washed and frozen at −80˚C for at least 12 hr. Thawed cell pellets were re-suspended in lysis buffer (50 mM Tris-HCl pH 7.5, 150 mM NaCl, 0.1% Tween 20, 1 mM β-mercapto ethanol, protease inhibitors) and lysed by sonication using four 1 min pulses at 40% output. Extracts were clarified using two rounds of centrifugation (20,000 X g, 20 min) and then bound to 1 ml of Ni-NTA beads in lysis buffer containing 5 mM imidazole. Beads were washed with lysis buffer containing 20 mM imidazole and eluted with lysis buffer containing 300 mM imidazole. Purified calcineurin heterodimer were dialyzed in buffer (50 mM Tris-HCl pH 7.5, 150 mM NaCl, 1 mM β-mercapto ethanol) and stored in 10–15% glycerol at −80˚C.

## Bead and antibody imaging

MRBLEs were imaged by transferring 20 μL of suspended beads (entire volume) onto a quartz microscope slide (Electron Microscopy Sciences, quartz microscope slide, 75 mm x 25 mm, 1 mm thick, cat. #72250–03), placing an additional quartz coverslip (Electron Microscopy Sciences, quartz

coverslip, 25 mm x 25 mm, 1 mm thick, cat. #72256–02) onto the droplet, and then depositing mineral oil around the edges of the coverslip to prevent the sample from drying out during imaging. MRBLEs were imaged largely as described previously (*Nguyen et al., 2017a*) in 11 channels: a bright field channel, a Cy5 fluorescence channel (using a SOLA light engine for excitation), and nine additional lanthanide channels (435, 474, 536, 546, 572, 620, 630, 650, 780 nm, with exposure times of 500, 1000, 500, 500, 375, 150, 75, 225, 2000 ms, respectively; Semrock, Rochester, NY) using excitation illumination generated by a Xenon arc lamp (Lambda SL, Sutter Instrument, Novato, CA) and directed through a 292/27 nm bandpass filter (Semrock, Rochester, NY) via a UV-liquid light guide (Sutter Instrument, Novato, CA) mounted in place of the condenser using a custom 3D printed holder.

## Image processing (code calling)

Bead images were processed using a custom-built open-source Python software package freely available through PyPI and GitHub and regularly maintained (*Harink et al., 2019*; *Harink, 2018*). Briefly, MRBLE boundaries were identified from bright field images, followed by quantification of median lanthanide intensities in each channel to identify embedded MRBLE codes. CN binding was then quantified for each MRBLE by calculating the median fluorescence intensity for bound protein associated with the outer shell of the MRBLE.

## Calcineurin binding assays (time series and dilution series)

Pre-incubation of CN with anti-His antibody significantly reduced observed background binding to sequences containing multiple basic residues due to cross-reactivity of anti-His antibody (*Figure 3— figure supplement 9*, *Figure 3—figure supplement 10*, *Figure 3—figure supplement 11*, *Figure 3—figure supplement 12*). Therefore, CN:$\alpha-$6xHis antibody-DyLight-650 complex (2.5 μM, CN:αHisAb650, Abcam ab117504) was prepared by pre-incubating 6x-His-tagged CN with equal concentration of αHisAb650 in CN buffer at ~5 ˚C for 1 hr. MRBLEs were prepared by transferring a 20 μL aliquot from a 600 μL suspension (~3000–8000 beads depending on pellet size) to a 100 μL PCR strip (20 μL for each CN concentration), exchanging buffer via 3 cycles of iterative pelleting, decanting, and resuspension with 0.1% PBST containing 5% BSA, pH = 7.5 (100 μL), and then left mixing at ~5 ˚C on a rotator in the same wash buffer overnight (~14 hr). MRBLEs were then buffer exchanged once again with CN binding buffer (50 mM Tris pH = 7.5, 150 mM NaCl, 0.1% TWEEN 20) via three iterative cycles of pelleting, decanting, and resuspension.

   To perform binding assays, the CN:αHisAb650 complex (at 2.5 μM) was serially diluted into tubes containing BSA-passivated MRBLEs and CN buffer (2 μM, 1μM, 500 nM, 250 nM, 125 nM, 62.5 nM) and incubated for 5 hr on a rotator at ~5 ˚C. After incubation, MRBLEs were pelleted, decanted, and washed once with 0.1% PBST (100 μL). After a final round of pelleting and decanting, 20 μL of 0.1% PBST was added and the beads were transferred to a quartz microscope slide for imaging. To confirm equilibrium conditions, we measured CN:MRBLE interactions for six peptides after incubation times ranging from 30 min to 24 hr. Binding appeared to reach equilibrium after ~5 hr (*Figure 2— figure supplement 1*), so all following experiments were performed after an incubation time of 5–6 hr.

## Affinity modeling (competitive binding assay)

The affinity ($K_d$) for each peptide in a MBRLE library was determined via a two-step nonlinear regression using a single-site binding model:

$$I_i = (I_{max}[CN_{total}])/(K_{d,i} + [CN_{total}]) \tag{1}$$

where $I_i$ is the median measured fluorescence intensity for each peptide, $K_{d,i}$ is the dissociation constant for that interaction, $[CN_{total}]$ is total CN concentration, and $I_{max}$ is a global variable representing the fluorescence intensity once MRBLE-pep beads are saturated. First, we determined the global saturation value ($I_{max}$), by globally fitting the top 80% highest-intensity MRBLE codes. Next, we globally fit the entire data set using this $I_{max}$ value to yield an estimated absolute affinity ($K_d$) for all peptides in the library. Given that the estimated peptide concentration in these assays (20 nM/x, where x represents the number of species probed) is significantly lower than estimated $K_d$ values of these

interactions, we make the approximation $[CN_{total}] \approx [CN_{free}]$. Differences in binding affinity were calculated relative to a reference peptide using the standard equation:

$$\Delta\Delta G = RTln(K_{d,i}/K_{d,ref}) \qquad (2)$$

To calibrate measured affinities across multiple assays, we first determined an absolute affinity for a single high-affinity reference peptide (MAGPHPVIVITGPHEE) using the mean of the 'Triplicate low' value (*Figure 2C*), 'Triplicate high' value (*Figure 2—figure supplement 2*), and value from a binding assay in which this peptide appeared alone (data not shown, $K_d$ = 980 nM). Next, we used this reference value and calculated differences in binding affinity ($\Delta\Delta G$) to estimate absolute $K_d$ values for each peptide within the calibration and full calibration libraries (*Figure 2—source data 1*, *Figure 2—source data 2*, *Figure 5—source data 1*, *Figure 5—source data 2*, *Figure 5—source data 3*, *Figure 5—source data 4*).

## Rosetta-based 'Sequence Tolerance' method

For estimating tolerated amino acid substitutions at each position in the two available CN-PxIxIT co-crystal structures with well-defined electron density (positions −1 to 9 of the PxIxIT motif) we used the generalized sequence tolerance module of the Rosetta Backrub server available at https://kortemmeweb.ucsf.edu/backrub (*Smith and Kortemme, 2011*). Briefly, the protocol samples amino acid residues in different rotameric conformations on backbone ensembles generated using Rosetta Backrub simulations and records low energy amino acid sequences, from which amino acid frequencies of tolerated substitutions are derived. In contrast to the published protocol, we modified only one position in the peptide at a time, to more closely mimic the experimental measurements. We used ensembles of 100 backbone structures with a *kT* value of 0.228; self-energies and interchain interaction energies were reweighted using the default scaling factors of 0.4 and 1, respectively.

## Rosetta-based method for estimating the energetic effects of amino acid substitutions ('flex_ ddG')

To estimate changes in binding energy upon mutation ($\Delta\Delta G$), we used the Rosetta flex_ddG protocol (*Barlow et al., 2018*). We systematically substituted all twenty natural amino acids at each position (−1:9) for the two peptides with available crystal structures bound to calcineurin (CN-PVIVIT, CN-IAIIIT), with the remaining positions restricted to the wild-type amino acid residues in each crystal structure. Briefly, the flex_ddG protocol uses the Rosetta Talaris 2014 energy function, minimization with harmonic restraints, Rosetta Backrub simulations to generate conformational ensembles, mutation and optimization of side chain conformations, and another final retrained minimization step. We followed the protocol published in *Barlow et al. (2018)*, except using 10,000 Backrub steps instead of the default value of 35,000 steps.

## FoldX calculations to estimate the energetic effects of amino acid substitutions

To prepare systems for analysis with FoldX, we obtained the PDB entries for calcineurin bound to a PVIVIT peptide (PDB entry 2P6B) and bound to a IAIIIT peptide (PDB entry 3LL8). We then deleted chains C and D, as well as accessory ions and waters, from both structures. In the 2P6B structure, we also removed residues Pro4, Glu16 and the -NH2 cap to match the resolved portions of the peptide present in the 3LL8 structure. We used two FoldX commands to perform the FoldX calculations. First, we ran the RepairPDB command on each system with default options to generate structures in a suitable format for the subsequent step. Second, for each mutation, we ran the BuildModel command for 10 iterations. We then extracted delta-delta Gs from the automatically generated output file that reported differences in the delta Gs between wild type and mutant. Data values in *Figure 4* correspond to median value of the delta-delta Gs for each of the ten iterations for a given mutant.

## *In vivo* calcineurin activity assay

PxIxIT peptides were fused to eGFP in pEGFPc1 vector. HEK293T cells were transfected with pEGFPc1 clones, pNFAT-Luc and CMV-Renilla in a 6-well plate format. 18 hr post transfection, 1μM FK506 or DMSO (vehicle) was added to the media as needed. 36 hr post transfection, cells were

treated with 1 mM Ionomycin and 1 mM phorbol 12,13 di-butyrate to activate calcineurin and AP-1 (via PKC) respectively. 6 hr after pathway activation, cells were collected, washed in PBS and re-suspended in DMEM media. 80% of the cells were used to measure luciferase activity and renilla using the Dual-Glo assay system (Promega) with three technical replicates. The remaining cells were frozen and stored at −80°C. Cell lysates were prepared in RIPA buffer. 15–20 µg of lysate was analyzed by Western for expression of GFP. GFP signal was normalized to either actin or tubulin. Luciferase activity (normalized to renilla expression) was further normalized to eGFP expression. Data represent at least three experimental replicates.

## Acknowledgements

This work was supported by NIH/NIGMS grants DP2GM123641 and R01GM107132 to PMF and R01GM119336 to MSC. In addition, PMF is a Chan Zuckerberg Biohub Investigator and acknowledges support from the Beckman Foundation, the Sloan Research Foundation, and PMF and MSC acknowledge the support of a joint Bio-X Interdisciplinary Initiatives Fund seed grant. TK is a Chan Biohub investigator and supported by NIH/NIGMS grants R01GM117189 and R01 GM110089. We thank Prof. Justin Kinney (Cold Spring Harbor Laboratory) for helpful discussions and software used to generate logos; we thank Prof. Dan Herschlag (Stanford) for essential feedback on the manuscript.

## Additional information

### Funding

| Funder | Grant reference number | Author |
|---|---|---|
| National Institute of General Medical Sciences | DP2GM123641 | Polly Morrell Fordyce |
| National Institute of General Medical Sciences | R01GM107132 | Kurt S Thorn |
| National Institute of General Medical Sciences | R01GM119336 | Martha S Cyert |
| National Institute of General Medical Sciences | R01GM117189 | Tanja Kortemme |
| National Institute of General Medical Sciences | R01GM110089 | Tanja Kortemme |
| Chan Zuckerberg Biohub | | Tanja Kortemme Polly Morrell Fordyce |
| Alfred P. Sloan Foundation | | Polly Morrell Fordyce |
| Arnold and Mabel Beckman Foundation | | Polly Morrell Fordyce |

The funders had no role in study design, data collection and interpretation, or the decision to submit the work for publication.

### Author contributions

Huy Quoc Nguyen, Conceptualization, Resources, Formal analysis, Investigation, Visualization, Methodology, Writing—original draft, Writing—review and editing; Jagoree Roy, Resources, Investigation, Writing—original draft, Writing—review and editing; Björn Harink, Data curation, Software, Formal analysis, Visualization; Nikhil P Damle, Conceptualization, Formal analysis, Investigation; Naomi R Latorraca, Formal analysis, Performed all FoldX analysis; Brian C Baxter, Resources, Synthesized all lanthanide nanophosphors; Kara Brower, Resources, Writing—review and editing; Scott A Longwell, Resources, Expressed and purified tagged calcineurin; Tanja Kortemme, Software, Supervision, Writing—review and editing; Kurt S Thorn, Conceptualization, Formal analysis, Supervision, Project administration; Martha S Cyert, Conceptualization, Supervision, Funding acquisition, Writing—original draft, Project administration, Writing—review and editing;

Polly Morrell Fordyce, Conceptualization, Formal analysis, Supervision, Funding acquisition, Visualization, Writing—original draft, Project administration, Writing—review and editing

## Author ORCIDs

Huy Quoc Nguyen (iD) https://orcid.org/0000-0002-5252-0062
Martha S Cyert (iD) http://orcid.org/0000-0003-3825-7437
Polly Morrell Fordyce (iD) https://orcid.org/0000-0002-9505-0638

## Decision letter and Author response

Decision letter https://doi.org/10.7554/eLife.40499.085
Author response https://doi.org/10.7554/eLife.40499.086

# Additional files

## Supplementary files

• Supplementary file 1. List of literature affinities and references.
DOI: https://doi.org/10.7554/eLife.40499.079

• Supplementary file 2. Calculated savings and references vs other techniques.
DOI: https://doi.org/10.7554/eLife.40499.080

• Transparent reporting form
DOI: https://doi.org/10.7554/eLife.40499.081

## Data availability

All data generated or analysed during this study are included in the manuscript and supporting files. In addition, all data generated or analyzed during this study are available in an associated public OSF repository (https://doi.org/10.17605/OSF.IO/FPVE2).

The following dataset was generated:

| Author(s) | Year | Dataset title | Dataset URL | Database and Identifier |
|---|---|---|---|---|
| Polly Morrell Fordyce, Huy Quoc Nguyen, Björn Harink | 2018 | Quantitative mapping of protein-peptide affinity landscapes using spectrally encoded beads | https://doi.org/10.17605/OSF.IO/FPVE2 | Open Science Framework, 10.17605/OSF.IO/FPVE2 |

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
