## [Decision Letter]

Thank you for submitting your work entitled "Quantitative mapping of protein-peptide affinity landscapes using spectrally encoded beads" for consideration by *eLife*. Your article has been reviewed by two peer reviewers, and the evaluation has been overseen by a Reviewing Editor and a Senior Editor. The following individual involved in review of your submission has agreed to reveal his identity: Toby Gibson (Reviewer #2).

Our decision has been reached after consultation within the editorial board. Based on these discussions and the individual reviews below, we regret to inform you that your work will not be considered further for publication in *eLife*. In particular, there was uniform agreement that the issues raised by reviewer 1 are highly unlikely to be addressed in revision.

*Reviewer #1:*

In this study, the authors present a new technique to measure the affinity of protein-peptide interaction in parallel. They apply their method to the well-known Calcineurin-PxIxIT interaction, obtain more data on this interaction, and use it to identify high-affinity peptides. They validate some of these peptides in a cell-based assay.

This work is based around the bead-encoded peptide-synthesis and subsequent affinity measurements using the spectral encoding. This all strikes me as technically state-of-the-art and quite elegant. However, from the paper as written, it is a bit difficult to get a true sense of the advantages of the new method over existing ones, and no direct comparison is attempted in the manuscript. For instance, mapping of the affinities of substitutions (as in Figure 4) surely would be possible using SPOT (to a slightly different degree, also using Phage). The shown r^2^ of 0.68 is pretty good, but not all that much higher than what has been claimed for SPOT (though a little higher than for phage – see e.g., Ivarsson et al., where, of course, a much higher number of peptides is measured). One should also note that this is for the same protein – incidentally, modern computational methods (such as alchemy-based ones, e.g., TI or CGI) achieve these sort of correlations, and usually perform better than rosetta. As high-quality structures are available, they would be applicable in this case, and may perform similar to the new bead-based method.

Similar things hold for the high-affinity variants obtained here (Figure 5 or later). While obtaining low-nanomolar binding peptides is really quite good, this domain is (as the authors themselves state) what phage display really is optimized for. While their new method can incorporate non-canonical amino acids, that is similarly true of novel synthetic library methods as what recently came out of the Pentelute lab.

Finally, while the authors seem to emphasize ability to detect "weak" affinities, I only see convincing data for affinities of up to maybe 50uM, which is all still detectable using techniques such as AP-MS or phage.

I should note that I do not really want to sound too negative. I see a new technique that brings substantial technical innovation that could be very exciting in this field. However, I find it difficult to ascertain its advantages over the variety of existing methods in the manuscript as presented. I would strongly suggest to include more direct comparisons with existing methods (such as SPOT, ProP-PD, hold-up, etc.) as well a more detailed discussion thereof.

A few additional points:

1) The authors state "CN binds PxIxIT peptides with weak affinities.… estimates for known substrates have varied over a wide range.…". These statements are somewhat contradictory in itself and, in any case, the affinities listed in Figure 2A seem to be all in the single digit uM, which is neither a wide range, nor particularly weak (roughly the expected range for domain-motif interactions).

2) Given the relatively length of the current paper, it would be good scholarship to discuss existing methods (including the variety of phage-display approaches, SPOT arrays and the recently developed holdup assay) – currently, this all happens in part of a single paragraph, with some of the methods I point out above not even mentioned.

3) In a similar vein, the citations (or lack thereof) are a bit puzzling. A 12-year old paper (Neduva and Russell) is cited regarding motif-interactions in Y2H/AP-MS, when most of the relevant work was published in the last decade. For Proteomic phage display, only a review paper is cited.

*Reviewer #2:*

This paper introduces MRBLE-pep, a medium-scale throughput protein-peptide affinity tool and applies it to calcineurin and variants of the PxIxIT docking motif. Using MRBLEs – colour-coded microbeads, each position in the original peptide can be substituted with all other amino acids. Systematic evaluation of binding determinants can then be rapidly determined using small amounts of reagents. Efficient methods for examining SLiM specificities are vey much needed given the abundance of protein:motif interactions in cell regulation and therefore MRBLE could see widespread adoption. Although the PxIxIT motif is well known from several calcineurin substrates, there has been uncertainty regarding the motif specificity determinants as well as the cooperative involvement of the second LxVP docking motif.

Weight matrices (PSSMs/HMMs) are often used to represent SLiMs in sequence searches but assume that each position in the sequence can be treated independently of the others. Regular expressions are also in widespread use but are overdetermined and insufficiently flexible regarding sequence space. The results here can be used to improve bioinformatic PxIxIT detection in candidate substrate proteins. However, the results also emphasise that positions in the PxIxIT motif are not independent and also that flanking residues affect the binding affinity (which has been seen for a number of other linear motifs too). The data from MRBLE-pep could allow a different computational approach whereby candidate motifs in protein sequences could be compared to and ranked by the total landscape affinity data.

No substantive concerns were identified.

[Editors’ note: what now follows is the decision letter after the authors submitted for further consideration.]

Thank you for resubmitting your work entitled "Quantitative mapping of protein-peptide affinity landscapes using spectrally encoded beads" for further consideration at *eLife*. Your article has been reviewed by two peer reviewers, and the evaluation has been overseen by a Reviewing Editor and a Senior Editor.

The manuscript has been improved but there are some remaining issues that need to be addressed before acceptance, as outlined below in the first review below. Normally, we try to avoid second revisions, but we feel reviewer #1's comments should be addressed and we hope you will be able to do this easily and quickly.

*Reviewer #1:*

Essentially, the revisions by the authors haven't changed my opinions drastically. I still see this is as an innovative technology that will likely be of interest; however, the paper is still hampered by some shortcomings that I think could be improved pretty easily. In a way I find it a bit frustrating that the authors choose to write a full two pages of rebuttal to a simple point (correlations to measured affinity - compare these R^2^s between SPOT and MRBLE-pep) when they could have simply provided two numbers. Also, little of these two pages of rebuttal seems to have made it to the text (or was in there to begin with). Why not provide some kind of comparison of correlation to measured affinity to SPOT in the paper? I'm reasonably certain MRBLE-pep would come out on top.

Similarly, the authors add calculations using FoldX (that I didn't ask for). Not sure why they include comparisons of FoldX to Rosetta in the main text/figure (these would be of interest in computational papers and indeed have been done a bit, but I doubt the main readership of this paper will be interested). Also, I find it a bit frustrating that while they added a many new figures, the one figure that I think would have added to the paper (again a simple direct comparison of ddGs derived from MRBLE-pep to ones derived from rosetta OR FoldX vs. measured ddGs, i.e. combining the panels of Figure 4—figure supplement 4 into one panel and comparing it to the appropriate one for MRLBLE-pep). It's fine if the authors don't want to do any TI/FEP calculations (again, this isn't a computational paper), though if they do want to make the point that FoldX/Rosetta don't perform well on solvent exposed residues (not surprising), they should also mention that this is something that TI/FEP would, at least in theory, do better at.

They did add a much more appropriate introduction with discussion of the literature.

*Reviewer #2:*

The original MRBLE-pep manuscript was rather let down by superficial treatment of other PPI methods and how MARBL-pep compares to them. The resubmitted manuscript is greatly improved in this regard. This is important because SLiM researchers need to be able to understand which methods are suitable for the projects they have in mind. These range from the low throughput "gold standard" ITC up to whole proteome screens of the "disorderome" with phage display to identify novel SLiM candidates. I believe that MARBL-pep will see significant adoption and will help to define many SLiM motif patterns. The data obtained for the chosen target, the Calcineurin-binding PxIxIT motif, will hopefully now be applied in the identification of novel substrates for this medically important phosphatase. I have no further concerns.

---

## [Author Response]

[Editors’ note: the author responses to the first round of peer review follow.]

We are writing to follow up on the recent *eLife* review of our manuscript “Quantitative mapping of protein-peptide affinity landscapes using spectrally encoded beads”. We believe that the criticisms are readily addressed and stem largely from the fact that we didn't clearly explain critical differences between our novel assay and current approaches. To frame this discussion, we provide an overview of the reviewers' comments below and attach a detailed response to each of the comments and suggestions.

Reviewer #2 (Toby Gibson) was very positive, noting that “efficient methods for examining SLiM specificities are very much needed given the abundance of protein:motif interactions in cell regulation”, that “MRBLE could see widespread adoption”, and that “the data from MRBLE-pep could allow a different computational approach whereby candidate motifs in protein sequences could be compared to and ranked by the total landscape affinity data”. We note that Toby Gibson is a world expert in protein-SLiM interactions, and are flattered by his positive reception to the work.

Reviewer #1 stated that the work was “technically state-of-the-art and quite elegant” and that they “see a new technique that brings substantial technical innovation that could be very exciting in this field”. However, they state that they found it “difficult to ascertain its advantages over the variety of existing methods in the manuscript as presented” and would “strongly suggest to include more direct comparisons with existing methods (such as SPOT, ProP-PD, hold-up, etc)”. This helpful suggestion is easily addressed in the text of the manuscript. To briefly summarize advantages of MRBLE-pep:

1) MRBLEs have been engineered to have very slow on- and off-rates (half-lives of ~ 5 hours), allowing equilibrium measurement of even weak and transient SLiM interactions to yield quantitative thermodynamic constants (K_D and ΔΔG). This is simply not possible with SPOT arrays or ProP-PD, as off-rates for domain-SLiM interactions are on the order of seconds, much shorter than typical washing steps such that interactions with fast off-rates are preferentially lost.

2) MRBLE-pep incorporates in-line quality control to provide critical negative information about residues that ablate or reduce binding that is not available via SPOT or ProP-PD.

3) MRBLE-pep allows direct assessment of the effects of post-translational modifications (not possible with ProP-PD).

4) MRBLE-pep requires significantly less material than current low-throughput quantitative techniques (~60000x less protein and ~6000x less peptide than micro/nano ITC).

Finally, we note that we apply this new technology towards studying a system of great biological interest and reveal new information about determinants of specificity. As Toby Gibson notes, “the results here could be used to improve bioinformatic PxIxIT in candidate substrate proteins” and “emphasize that positions in the PxIxIT motif are not independent and also that flanking residues affect the binding affinity”.

Reviewer #1:

In this study, the authors present a new technique to measure the affinity of protein-peptide interaction in parallel. They apply their method to the well-known Calcineurin-PxIxIT interaction, obtain more data on this interaction, and use it to identify high-affinity peptides. They validate some of these peptides in a cell-based assay.This work is based around the bead-encoded peptide-synthesis and subsequent affinity measurements using the spectral encoding. This all strikes me as technically state-of-the-art and quite elegant.

We appreciate the reviewer’s kind opinion that the method is “technically state-of-the-art and quite elegant”!

However, from the paper as written, it is a bit difficult to get a true sense of the advantages of the new method over existing ones, and no direct comparison is attempted in the manuscript. For instance, mapping of the affinities of substitutions (as in Figure 4) surely would be possible using SPOT (to a slightly different degree, also using Phage). The shown r^2^ of 0.68 is pretty good, but not all that much higher than what has been claimed for SPOT (though a little higher than for phage – see e.g., Ivarsson et al., where, of course, a much higher number of peptides is measured).

The reviewer makes an excellent point. We did not clearly explain the advantages and disadvantages of MRBLE-pep relative to current techniques, and we also did not emphasize why quantitative measurements of interaction affinities are so important! We appreciate the helpful suggestions for how to improve the manuscript. In response, we have extensively revised the text in the Introduction and Discussion to include several paragraphs comparing and contrasting MRBLE-pep to current technologies.

To ensure that we clearly communicate these points, we also discuss the advantages and disadvantages of these approaches here, where we are not constrained by length.

Array-based methods:

Array-based methods provide the ability to probe for interactions between a protein of interest and 10s^-1^00s (SPOT arrays) or up to millions (ultra-high density arrays) peptides in a single experiment (Frank et al., 1990; Atwater et al., 2018; Buus et al., 2012; Forsstrom et al., 2014; Price et al., 2012; Carmona et al., 2015). In most cases, peptides are synthesized in situ via chemical synthesis approaches, making it possible to profile interactions between proteins and peptides containing unnatural amino acids or post-translational modifications at known positions. However, array-based methods cannot return quantitative interaction affinities (dissociation constants (K_D_s) or relative affinities (ΔΔGs)) because the measurements do not take place at thermodynamic equilibrium. Following incubation of proteins of interest with a peptide of interest, arrays must be washed extensively prior to imaging to quantify bound material; when a fluorescently-labeled antibody is used to detect bound protein, two incubations and two washing steps are required. In either case, the long washing times (minutes to hours) relative to the interaction dissociation rates (seconds) (Zhou et al., 2012; Dogan et al., 2015; and discussion in Ivarsson et al., 2019) mean that only the most persistent interactions are retained and interactions with faster off-rates are preferentially lost. Multiple comparisons between bound protein intensities and measured affinities for array-based experiments have established that while intensities typically correlate well with affinities for high-affinity interactions, arrays fail to recover weaker interactions. Finally, array-based methods are associated with high numbers of false positive and false negative measurements because it is difficult to evaluate the yield and purity of peptides present in each spot (Blikstad et al., 2015).

MRBLE-pep offers the following advantages relative to array-based methods:

1) Equilibrium measurements to yield affinities. As shown in Figure 2—figure supplements 1 and 2, we have deliberately engineered MRBLE polymer beads to have extremely slow on- and off-rates. This property makes it possible to remove unbound material and image to quantify bound protein with the washing steps taking place over a period of time that is much smaller than the half-life of the reactions, thereby approximating equilibrium and allowing direct measurement of interaction affinities.

2) In-line quality control for yield negative binding information. The use of different chemical linkers in the MRBLE core and periphery make it possible to cleave synthesized peptides from the MRBLE core only and directly assess their quality via MALDI-TOF mass spectrometry while preserving peptides presented at the periphery for binding measurements. If quality control reveals issues with synthesis, particular MRBLE-peptide combinations can simply be re-synthesized, re-tested, and added to the pool.

3) Statistical replicates. Each pooled measurement assesses binding of protein to ~ 30 beads containing the same embedded code and displayed sequence, allowing robust discrimination of even subtle differences in binding between sequences.

Display-based methods: Display-based methods have the advantage of profiling interactions between SLiM-binding proteins and large numbers (up to 10^10^) of candidate SLiMs. While approaches using combinatorial peptide libraries were biased towards identifying hydrophobic substrates, Ylva Ivarsson’s lab has pioneered new approaches for profiling interactions between SLiM-binding proteins and peptides tiled across intrinsically disordered regions of the proteome (Pro- PD) that remove this bias, as well as a newer approach that profiles peptides containing either serines and threonines or phosphomimetic residues at the same position to examine phosphorylation-dependent binding specificities.

However, a primary disadvantage of display-based methods is that they cannot return thermodynamic interaction affinities. Display-based techniques require multiple rounds of selection and washing, and as a result, take place out of equilibrium and provide only semi-quantitative estimates of binding strength biased towards SLiMs with the slowest dissociation rates. While computational approaches for extracting binding affinities from all rounds of selection have recently been developed for and applied towards high-throughput measurements of transcription factor-DNA affinities via SELEX approaches (e.g. No Read Left Behind, Rastogi et al., PNAS 2018), current Pro-PD and ProP-PD experiments typically consider NGS reads from a single round of enrichment (e.g. the fourth round of selection) only. In addition, display-based techniques cannot return information about the effects of post-translational modifications. While ProP-PD incorporates phosphomimetic residues in place of native serines and threonines to attempt to probe the effects of phosphorylation, we demonstrate here that phosphomimetic residues do not always fully mimic the effects of phosphorylation (here, phosphorylation at position 9 increases affinities 50-fold while phosphomimetic residues have essentially no effect). Finally, display-based techniques return a high number of false positives and false negatives and must therefore be validated via an alternate approach such as isothermal calorimetry. Finally, all selection-based methods, including Pro-PD, return only positive information (i.e. identify motifs that interact with the domain in question), but do not directly test if a particular sequence binds. Therefore, critical information about residues that prevent binding is not identified.

In fact, MRBLE-pep is a perfect complement to Pro-PD. After identifying selected sequences via Pro-PD, MRBLE-pep could be used to validate binding to identified peptides, determine binding affinities and test effects of amino substitutions on affinity for comprehensive motif characterization. We have modified the manuscript to make this point more explicit, and appreciate the reviewer’s suggestion.

Other methods:

Y2H and AP-MS: A recent large-scale yeast-two-hybrid (Y2H) screen identified 14000 potential binary protein-protein interactions to reveal the human interactome network, representing a milestone in knowledge of this network (Rolland et al., 2014). Over the following years, other groups have expanded the knowledge of this network via AP-MS assays to detect interactions (Huttlin et al., 2015; Huttlin et al., 2017), revealing ~23,000 and 56,000 interactions, respectively. However, interactions between SLiMs and SLiM-binding proteins remain particularly underrepresented (Van Roey et al., 2014; Davey et al., 2017). Underrepresentation in AP-MS is likely due to the low affinities and fast dissociation rates for SLiM-mediated interactions and the need for washing steps in this technique. In addition, neither Y2H or AP-MS methods claim to return quantitative interaction affinities. Moreover, AP-MS reports only on the presence of an interaction between 2 protein components, without information about whether this interaction is binary or what protein fragment/motif is responsible.

Hold-up assay: The hold-up assay represents the most direct competitor to the work presented here. This method was first described in an elegant and thorough Nature Methods paper from 2015 by Vincentelli and colleagues in which they profiled interactions between 210 PDZ-peptide pairs with known affinities and then mapped the specificity landscape between 2 viral PDZ-binding motifs from human papillomavirus E6 oncoproteins and 209 PDZ domains. While this assay detects complexes binding at equilibrium (in contrast to other high-throughput methods mentioned above), in its standard form, the assay returns only ranked lists of likely interaction strengths (not actual affinities). Deriving quantitative affinity information for complexes with K_D_ > 8 µM requires repetition of the assay at higher protein concentrations in order to ensure that available soluble protein is not significantly depleted within the unbound fraction. As a result, the material requirements for the hold-up assay significantly exceed those for the assay presented here. Finally, we note that the hold- up assay has never been cited in subsequent experimental work, suggesting that there are significant technical barriers to widespread adoption.

Because most high-throughput technologies take place out of thermodynamic equilibrium and therefore cannot measure quantitative interaction affinities (equilibrium constants), the vast majority of SLiMs have been characterized by low- throughput experimentation (see Davey et al., 2017; Dinkel et al., 2016; Toniian et al., 2008). Rather than viewing high-throughput screening methods as competing technologies for MRBLE-pep, we believe that the two approaches are highly complementary. The large sequence spaces accessible to display-based methods make it possible to scan large sequence spaces for candidate interaction motifs. After such a scan, MRBLE-pep can be used to: (1) quickly and efficiently scan through thousands of potential SLiMs to validate true hits, (2) directly measure SLiM interaction affinities, and then (3) systematically perturb identified motifs to map the surrounding binding affinity landscape, assess binding specificity, and determine a position specific affinity matrix that can be used to more accurately predict candidate downstream targets across the proteome. MRBLE-pep has the potential to streamline tedious downstream validation and affinity measurement process relative to traditional quantitative biotechniques such as fluorescence polarization, surface plasmon resonance, and isothermal calorimetry. We have amended the Discussion to more clearly emphasize that we view MRBLE-pep as a method that is complementary to (rather than competitive with) high-throughput discovery techniques, and hope that this addresses this reviewer’s concerns.

One should also note that this is for the same protein – incidentally, modern computational methods (such as alchemy-based ones, e.g., TI or CGI) achieve these sort of correlations, and usually perform better than rosetta. As high-quality structures are available, they would be applicable in this case, and may perform similar to the new bead-based method.

We agree with the reviewer that including other computational methods will likely improve the ability to accurately predict the energetic effects of SLiM mutations. In particular, the reviewer suggests that alchemical free energy calculations may improve these predictions. However, a previous paper (Aldeghi et al., 2018) directly compared performance between these calculations and Rosetta flex_ddG and concluded:

“We show that both the free energy calculations and Rosetta are able to quantitatively predict changes in ligand binding affinity upon protein mutations, yet the best predictions are the result of combining the estimates of both methods.”

They also note the difference in computational cost:

“It is important to put the performances of the calculations in the context of their computational cost. With the free energy calculations, each ΔΔG estimate took between 2 and 5 days on a node with 20 CPU threads and 1 GPU (Intel Xeon E5-2630 v4; GTX 1080 Ti), depending on the size of the system (from ~30000 to ~100000 atoms). With the flex_ddG protocol, each ΔΔG estimate took up to a day on a single CPU core.”

To address the reviewer’s suggestion, we worked with Naomi Latorroca from Ron Dror’s lab at Stanford to use the FoldX server to predict the likely change in binding energy upon mutating different residues within PxIxIT peptides on calcineurin- PxIxIT binding affinities (see Figure 4 and associated supplementary figures and data sets). In many cases, the FoldX predictions outperformed those of the Rosetta flex_ddG protocol when assessing the ability to classify predicted substitutions as either stabilizing (ΔΔG < 0) or destabilizing (ΔΔG > 0), and we think that the manuscript has been improved as a result. We note, however, that the prediction accuracy remains somewhat low, consistent with the prevailing sentiment that predicting mutational effects for protein-peptide interactions remains a difficult problem due to the large number of possible conformations of the peptide when free in solution. MRBLE-pep therefore provides a unique opportunity to iterate quickly between computational predictions and experimental measurements, as high-throughput peptide synthesizers allow synthesis of up to 576 peptides on MRBLEs in parallel in about a week and downstream binding assays require only a few additional days. The inclusion of this analysis has substantially improved the paper and we appreciate the reviewer’s suggestion!

Similar things hold for the high-affinity variants obtained here (Figure 5 or later). While obtaining low-nanomolar binding peptides is really quite good, this domain is (as the authors themselves state) what phage display really is optimized for.

We appreciate the reviewer’s point that identifying super-binders is easily accomplished via a variety of other methods, including phage display, that allow searching of extremely large combinatorial libraries (up to 1010). However, the current approach demonstrated the ability to identify peptides with affinities 2 orders of magnitude stronger than those previously identified via phage display using only 400 measurements. This brings up an interesting question: what is the most efficient way to scan a potential sequence space? For peptides that bind as linear motifs, the results presented here suggest that systematically testing a single mutation at each position and then combining those results may provide an alternative method to obtain the same results via a much smaller number of experimental measurements.

While their new method can incorporate non-canonical amino acids, that is similarly true of novel synthetic library methods as what recently came out of the Pentelute lab.

We appreciate the discussion of the exciting new technologies that have emerged from the Pentelute lab. We regret failing to cite these in our initial submission and have amended this by including several citations in the current text. Again, however, we emphasize that these technologies are screening technologies that, while able to test a wide variety of potential interactions (including a wide variety of unnatural and modified amino acids), do not return quantitative interaction affinities. As with phage display, we view the current MRBLE-pep technology as complementary to, rather than competitive with, these screening technologies. We have amended the text to include references to 2 recent relevant publications from the Pentelute lab.

Finally, while the authors seem to emphasize ability to detect "weak" affinities, I only see convincing data for affinities of up to maybe 50uM, which is all still detectable using techniques such as AP-MS or phage.

We respectfully disagree with the reviewer’s suggestion that affinities of up to 50 µM are detectable using AP-MS, as prior analyses have demonstrated that low-affinity SLiM-mediated interactions are significantly underrepresented within these datasets, as described above. However, even if these interactions were not underrepresented within AP-MS datasets, there would still be a need to directly and quantitatively measure the interaction affinities to predict biological responses and downstream outputs, as well as to acquire *in vitro* data to distinguish between direct and indirect binding interactions. For this reason, we again suggest that MRBLE-pep may represent an ideal companion technology to quantify affinities for candidate AP-MS interactions to decode cellular signaling interaction networks and for use in whole-cell modeling efforts.

I should note that I do not really want to sound too negative. I see a new technique that brings substantial technical innovation that could be very exciting in this field. However, I find it difficult to ascertain its advantages over the variety of existing methods in the manuscript as presented. I would strongly suggest to include more direct comparisons with existing methods (such as SPOT, ProP-PD, hold-up, etc.) as well a more detailed discussion thereof.

We appreciate the reviewer’s encouraging comment that the review is not meant to sound too negative, as well as the sentiment that this is “a new technique that brings substantial technical innovation that could be very exciting in this field”! We believe that the reviewer’s suggestions to include more direct comparisons between MRBLE-pep and these existing techniques as well as to explore additional computational methods for predicting mutational effects on protein-peptide affinities have strengthened the paper considerably. We have significantly revised the Introduction in response to these helpful suggestions.

A few additional points:1) The authors state "CN binds PxIxIT peptides with weak affinities.… estimates for known substrates have varied over a wide range.…". These statements are somewhat contradictory in itself and, in any case, the affinities listed in Figure 2A seem to be all in the single digit uM, which is neither a wide range, nor particularly weak (roughly the expected range for domain-motif interactions).

We agree with the reviewer that single digit µM Kd values are typical for SLiM-mediated binding interactions. However, SLiM-mediated binding interactions are typically referred to as moderate-to-weak affinity and transient (for example, please see Davey et al., 2012). To address this suggestion, we have amended the text to read “CN binds PxIxIT peptides with moderate-to-weak affinities…”. When we suggest that estimates for known substrates have varied over a wide range, we mean to say that published estimates for the same specific interaction between CN and a particular SLiM have varied over a wide range. To clarify this, we have amended the text to read: “… prior literature affinity estimates for a given substrate have varied over 50-fold.”

2) Given the relatively length of the current paper, it would be good scholarship to discuss existing methods (including the variety of phage-display approaches, SPOT arrays and the recently developed holdup assay) – currently, this all happens in part of a single paragraph, with some of the methods I point out above not even mentioned.

We appreciate the suggestion and have revised the Introduction extensively to include a more detailed discussion of other methods, as detailed above. We sincerely regret the oversight in the initial submission.

3) In a similar vein, the citations (or lack thereof) are a bit puzzling. A 12-year old paper (Neduva and Russell) is cited regarding motif-interactions in Y2H/AP-MS, when most of the relevant work was published in the last decade. For Proteomic phage display, only a review paper is cited.

The reviewer makes an excellent point. In re-reading the paper, it’s clear that we were overly focused on presenting the data and did an embarrassingly poor job of referencing much of the recent literature. We again regret this oversight and have extensively rewritten the Introduction and amended the paper to include a much more comprehensive discussion of recently published work. This is another case where we feel that the reviewer’s suggestions have significantly improved the paper by making comparisons to other methods more clear!

Reviewer #2:

[…] Weight matrices (PSSMs/HMMs) are often used to represent SLiMs in sequence searches but assume that each position in the sequence can be treated independently of the others. Regular expressions are also in widespread use but are overdetermined and insufficiently flexible regarding sequence space. The results here can be used to improve bioinformatic PxIxIT detection in candidate substrate proteins. However, the results also emphasise that positions in the PxIxIT motif are not independent and also that flanking residues affect the binding affinity (which has been seen for a number of other linear motifs too). The data from MRBLE-pep could allow a different computational approach whereby candidate motifs in protein sequences could be compared to and ranked by the total landscape affinity data.No substantive concerns were identified.

We appreciate the reviewer’s kind words and positive assessment of the work! We particularly appreciate that the reviewer correctly noted the ability of MRBLE-pep to provide quantitative information about interaction epistasis. We have added an additional sentence to the manuscript to emphasize this point.

[Editors’ note: the author responses to the re-review follow.]

The manuscript has been improved but there are some remaining issues that need to be addressed before acceptance, as outlined below in the first review below. Normally, we try to avoid second revisions, but we feel reviewer #1's comments should be addressed and we hope you will be able to do this easily and quickly.

Reviewer #1:

Essentially, the revisions by the authors haven't changed my opinions drastically. I still see this is as an innovative technology that will likely be of interest; however, the paper is still hampered by some shortcomings that I think could be improved pretty easily. In a way I find it a bit frustrating that the authors choose to write a full two pages of rebuttal to a simple point (correlations to measured affinity – compare these R^2^s between SPOT and MRBLE-pep) when they could have simply provided two numbers. Also, little of these two pages of rebuttal seems to have made it to the text (or was in there to begin with). Why not provide some kind of comparison of correlation to measured affinity to SPOT in the paper? I'm reasonably certain MRBLE-pep would come out on top.

The reviewer makes an excellent suggestion, and we fully agree that this would be a very interesting and worthwhile comparison. Unfortunately, this is simply not possible with current published data, largely due to the technical challenges associated with measuring quantitative binding affinities. Prior literature includes estimates of measured affinities for only a handful of calcineurin-PxIxIT interactions, all 8 of which we also measured via MRBLE-pep (Figure 2C). The strong agreement between MRBLE-pep affinities and these 8 prior measurements is shown in Figure 2D (r2 = 0.68), with only NFATc2 (PRIEIT) showing significant deviation. To clarify that this data set represents all known prior published affinities, we have modified a sentence in the “MRBLE-pep yield quantitative measurements of calcineurin binding affinities” to read: “These 10 peptides (the “Triplicate low” dataset) represent all calcineurinbinding SLiMs for which affinities have been measured, including the known NFATc1, NFATc2, AKAP79, and RCAN1 natural CN-interacting PxIxIT binding site (citations), a set of 5 PVIVIT peptide mutants previously characterized via competitive fluorescence polarization assays (citations), and a scrambled negative control sequence.”

These peptides have not been measured via SPOT arrays, preventing the desired comparison. The only published paper using arrays to quantify binding to calcineurin assessed binding of tiled calcineurin fragments to a single PxIxIT motif (the NFATc2 SLiM SGPSPRIEITPSH) and was unable to detect any binding between calcineurin fragments and the known PVIVIT substrate (Erdmann et al., ChemBioChem 2018). To fully address the reviewer’s request without relying on previously published literature, we attempted to perform ITC for a series of calcineurin-peptide interactions for approximately 3 months but were ultimately unsuccessful.

Similarly, the authors add calculations using FoldX (that I didn't ask for). Not sure why they include comparisons of FoldX to Rosetta in the main text/figure (these would be of interest in computational papers and indeed have been done a bit, but I doubt the main readership of this paper will be interested). Also, I find it a bit frustrating that while they added a many new figures, the one figure that I think would have added to the paper (again a simple direct comparison of ddGs derived from MRBLE-pep to ones derived from rosetta OR FoldX vs. measured ddGs, i.e. combining the panels of Figure 4—figure supplement 4 into one panel and comparing it to the appropriate one for MRLBLE-pep).

Figure 4—figure supplement 5 and Figure 4—figure supplement 6 compare MRBLE-pep measurements to the predictions of FoldX and the Rosetta flex_ddG protocol, respectively. Of the prior published measurements of calcineurin- PxIxIT affinities, only a single study considered how systematic substitutions within the PVIVIT site affect measured affinities by performing fluorescence polarization measurements in the presence of competing unlabeled peptides to estimate affinities (Li and Hogan, 2007). The FoldX and Rosetta computational analyses estimate effects on affinity for all possible single-site substitutions within the PVIVITGPH site, and therefore require orthogonal measurements of single-site substitutions for comparison. The Li and Hogan experiments measure relative affinities for only the wildtype PVIVIT peptide and 2 examples of single-site substitutions from the PVIVIT motif (PVIVVT and PVIVIN); the other measurement includes 2 amino acid mutations (PVIAVT).

The only possible comparison between MRBLE-pep, FoldX, and Rosetta and prior published results therefore assesses their performance on 2 WT-like peptides (PVIVIT and PVIVVT) and one weak affinity variant (PVIVIN), resulting in a 3-point comparison for which 2 points are experimentally identical. The MRBLE-pep measurements alone (and not the FoldX and Rosetta computational predictions) correctly recapitulate that: (1) PVIVIT and PVIVVT have the same affinities) and (2) the PVIVIN substitution substantially reduces binding.

In response to the reviewer suggestion, we have included this figure as Figure 4—figure supplement 7, and also added the following to the text: “Comparisons of MRBLE-pep measurements and Rosetta and FoldX predictions of the energetic effects of single site substitutions with orthogonal measurements are complicated by a lack of previously published affinity data. However, a direct comparison between MRBLE-pep measurements and computational predictions for 2 single-site substitutions in the PVIVIT motif reveal that only MRBLE-pep correctly discerns that a PVIVVT substitution has little effect while a PVIVIN substitution reduces binding.”

It's fine if the authors don't want to do any TI/FEP calculations (again, this isn't a computational paper), though if they do want to make the point that FoldX/Rosetta don't perform well on solvent exposed residues (not surprising), they should also mention that this is something that TI/FEP would, at least in theory, do better at.

We thank the reviewer for this suggestion and have added the following sentence to the section “Comparing experimental measurements with results from structure-based computational predictions”: “Thermodynamic Integration and Free Energy Perturbation (TI/FEP) methods would likely enhance the ability to predict effects of mutating solvent-exposed residues, but are significantly more computationally expensive (Kilburg and Gallichio, 2016; Gallichio and Levy, 2011).”

They did add a much more appropriate Introduction with Discussion of the literature.

We thank the reviewer for this suggestion, agree that the discussion of the literature is much improved, and apologize for the initial oversight.